# An Integrated Evaluation of the National Water Model (NWM) Height Above Nearest Drainage (HAND) Flood Mapping Methodology

J. Michael Johnson[1]*, Dinuke Munasinghe[2], Damilola Eyelade[1], Sagy Cohen[2]

[1]University of California, Santa Barbara, Santa Barbara, 93106, USA
2University of Alabama, Tuscaloosa, AL 35487

*Correspondence to*: J. Michael Johnson ([jmj00@ucsb.edu](mailto:jmj00@ucsb.edu))

**Abstract.** Flood maps are needed for emergency response, research, and planning. The Height Above Nearest Drainage (HAND) technique is a low-complexity, terrain-based approach for inundation mapping using elevation data, discharge-height relationships, and streamflow inputs. The recent operational capacities of the NOAA National Water Model (NWM) and pre-processed HAND products from the University of Texas offer an operational framework for real-time and forecast flood guidance across the United States. In this study, we evaluate the integrated National Water Model (NWM) Height Above Nearest Drainage (HAND) flood mapping approach using 28 remotely sensed inundation maps and 54 reach-level catchments. The results show the NWM-HAND method tends to underpredict inundated cells in 4th order and lower order reaches but does better with a slight tendency to over predict, in high order reaches. An evaluation of the roughness coefficient used in the derivation of synthetic rating curves suggests it is the most important parameter for correcting these errors. Persistent inaccuracies do occur when NWM streamflow predictions are substantially biased (>60% mean absolute error between NWM and observed streamflow) and in regions of low relief. Overall, the NWM-HAND method does not accurately capture inundated cells but is quite capable of highlight regions likely to be at risk in fourth order streams and higher. While NWM-HAND should be used with cautiously when identify flood boundaries or making decisions predicated of whether a cell is dry or wet, its applicability as a high-level guidance tool along larger rivers is noteworthy.

## 1. Introduction

Floods are one of the deadliest natural hazards in the United States. Over the last 30 years, floods have killed, on average, 86 people annually, increasing in the last 10 years to 95, and between 2015 and 2017 to more than 100 (Lam, 2018). While floods used to be associated with coastal zones, 8 of the 10 US states with the largest flooding disasters in the last decade were landlocked (Lightbody and Tompkins, 2018). Such patterns can be expected to continue as climate and land use pressures amplify (Hirabayashi et al., 2013; Yin et al., 2018). While flood damage cannot be eliminated, it can be mitigated. This is particularly true when populations are given advanced warning, have faith in the forecast accuracy, and are provided actionable intelligence (Johnson et. al 2016, 2018). When emergency responders are given maps showing where water is and where it might be in the future,

better choices can be made that systematically save time, energy, and resources (National Research Council, 2009). Combined, these challenges suggest a need to better understand and forecast flood
events across the entire United States (U.S.).

Since 2015, efforts towards producing real-time and future inundation forecasts for U.S. have resulted in the compilation of a 10-meter resolution Height Above Nearest Drainage (HAND) layer for the Continental United States (CONUS: Liu et al., 2018). The methodology for coupling streamflow
predictions from the National Water Model (NWM) and HAND was initiated as part of the National Flood Interoperability Experiment (NFIE) (Maidment, 2016). The methodology has since been enhanced and added to the National (NWC) US flood forecasting framework (NOAA National Water Center, 2018). The current objective of the NWM-HAND approach is rapid flood prediction for the purposes of disaster warning and guidance. Model accuracy should therefore be viewed in this context
and expectations should be tempered while recognizing the importance of having an operational, continental scale flood forecasting system.

A number of studies have shown that the HAND methodology can produce accurate inundation maps from a known streamflow and carefully created HAND layers (Afshari et al., 2018; Rennó et al., 2008;
Rodda, 2005; Zheng et al., 2018). There has not been, however, an exploration of how errors in the NWM-HAND methodology, and its individual components, might compound and propagate through flood simulation. In this paper we evaluate the integrated skill of NWM-HAND by comparing simulated events against an observational dataset at two scales, the floodplain, and the reach-level catchment.

## 2. Background

The goal of this paper is to study the combined skill of the National Water Model (NWM) and the current CONUS wide HAND layers to produce inundation maps suitable for emergency response and guidance. We will do this by comparing simulated flood extents to an archive of observed events. In this section we introduce the HAND methodology, the existing CONUS-wide datasets, the National Water Model, and the flood map repository.

**2.1 Height-Above-Nearest-Drainage (HAND) Method**

As the name implies, a Height Above Nearest Drainage (HAND) map contains the vertical distance between a location and its nearest stream. Producing a HAND map requires a digital elevation model (DEM) and a spatial representation of a region's river network. In the USA, these primarily come from the USGS National Elevation and Hydrography Datasets respectively (NED and NHD).
Each object in the NHD is described as a 'reach' and has three realizations including the (1) outflow point (2) the catchment polygon and (3) the flowpath polyline. Each realization refers to the same feature and is indexed by a unique common identifier (COMID). Figure 1A illustrates a rasterized object in all three realizations.

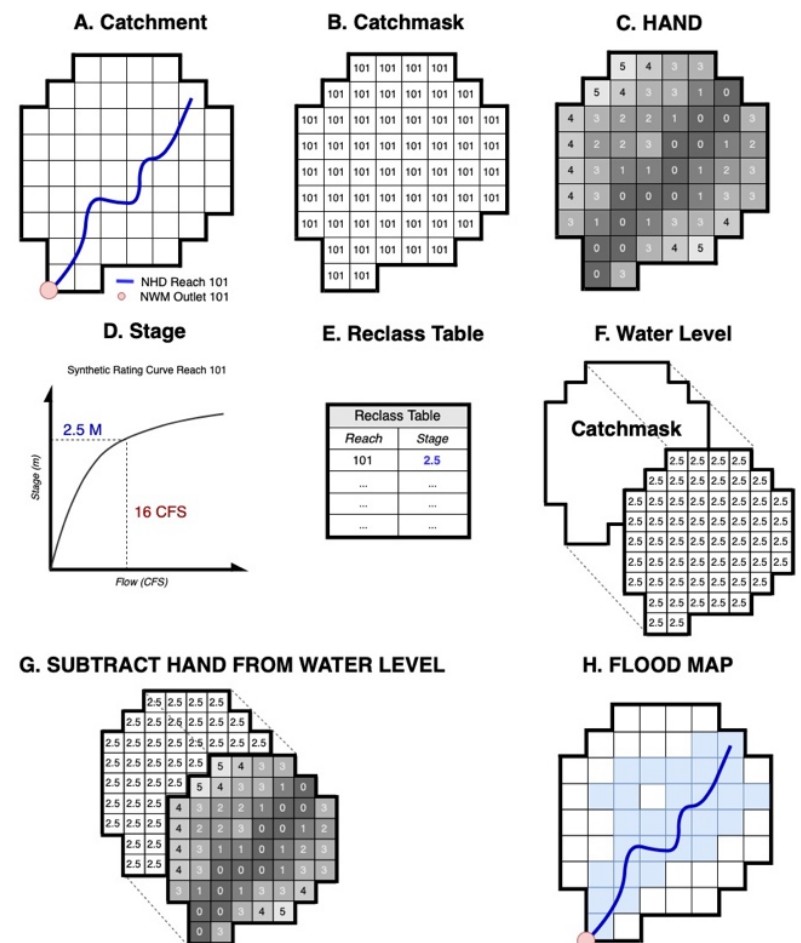

Figure 1: The HAND methodological workflow: (A) The contributing catchment to a defined outlet is rasterized to the resolution of the supporting DEM (B) This raster is reclassified to match the ID of the outlet. (C) In and out-of-stream cells are identified using the flowline vector. The relief between all out-of-stream cells from their nearest in-stream-cell is calculated to define a HAND raster. (D) From a given flow rate, a rating curve can be used to convert flow to stage. (E) A reclassification table can be built relating reach ID to stage. (F) The table can then be used to reclassify the catchmask, into a water-level raster. (G) Subtracting the HAND raster from the water-level raster yields a water-level above surface raster. All values less than 0 can be set to 0 and the remaining show the estimated flood height at each cell (H).

In the HAND methodology, a catchment mask, or catchmask, is created by rasterizing the catchment polygon to the DEM grid and assigning grid cell values equal to the COMID of the reach (e.g. 101) (Figure 1B). To generate a HAND raster, the flowpath is used to determine in-stream and out-of-stream cells. For all out-of-stream cells, the relief between that cell and the nearest in-stream cell is calculated and stored as the HAND cell value (Figure 1C).

Once computed, HAND raster's can be used to define hydraulic flood plain cross-sections. The derived geometries provide cross sectional areas (A) and hydraulic radius (R) inputs for the Manning's equation (Zheng et al., 2017, Eq. (1)):

$$Q(y) = \frac{1}{N} \times A \times R^{\frac{2}{3}} \times S^{\frac{1}{2}} \tag{1}$$

where Q(y) is the discharge for a given stage, N is the Manning's roughness coefficient, and S is the reach slope. From this equation, discharge (Q) requirements can be calculated for a range of stage values and stored as a synthetic rating curve (SRC; Zheng et al., 2017).

Given a flow value at a reach outlet (either observed or modelled) a stage can be generated from the SRC (Figure 1D). If more than one catchment is being modelled, the synthetic stage values can be stored according to COMID in a table (Figure 1E). Using this table, a new raster of water-depths can be created by reclassifying the catchmask raster (Figure 1F). Finally, subtracting the HAND raster from a water-depth raster yields an inundation map of above surface water depths (Figure 1G). Any value greater than zero indicates inundation and values less than zero indicate dry cells (Figure 1H).

## 2.2 HAND Products for the Continental United States

In 2017, HAND raster's and SRCs were generated for CONUS using the 10-meter NED and medium resolution NHD datasets on the ROGER supercomputing system at the University of Illinois Urbana Champaign (Liu et al., 2018; Zheng et al., 2017). In the pre-computed HAND rasters, relief was calculated via the TauDEM distance down function (Tesfa et al., 2011). In the SRC creation, reach slope was taken for the NHD attributes table, while a constant channel roughness (n) of 0.05 has been adopted for all of CONUS (Zheng et al., 2018, Johnson and Coll, 2017). All products are catalogued by HUC6 (the basin level units in the WBD) on the University of Texas (UT) Corral Server. (https://web.corral.tacc.utexas.edu/nfiedata/).

## 2.3 National Water Model (NWM)

The NWM serves as a cornerstone of the new NOAA Water Initiative to provide integrated predictive capabilities that promote resilience and mitigation of water risks (Graziano, 2017). In August 2016, version 1.0 was made operational, expanding the Nation's forecasting domain from approximately 9,000 gaged locations to 2.7 million reach outlets along the NHDPlusV2 (Gochis, 2018). The National Water Model uses the Noah Multi-Parameterization (Noah-MP) Land Surface Model (LSM) to simulate land surface processes across a 1km grid and a separate 250m grid to perform diffusive wave surface and saturated subsurface flow routing (Gochis, 2018). Once water is in channel, WRF-Hydro implements a standard 1-D Muskingum-Cunge (MC) hydrograph routing method using time varying parameter estimates.

Complimenting the operational products are 23-year reanalysis studies for NWM versions 1.0, 1.2, and 2.0. These products use downscaled NLDAS-2 climate forcing's with the standard NWM configuration. Unlike the operational Analysis and Assimilation product however, the reanalysis products do not assimilate observed streamflow and have been calibrated in limited number of basins (Gochis, 2016). In this study, the v1.2 reanalysis product was used (https://registry.opendata.aws/nwm-archive/).


**2.4 U.S. Flood Inundation Map Repository (USFIMR)**

Following the release of the NWM, academic partners at the University of Alabama developed the US Flood Inundation Map Repository (USFIMR) to provide inundation maps for past U.S. flood events.
These maps were derived using image classification techniques and a number of satellite sensors (e.g. Landsat, Sentinel-1, 2) with some ground truthing based on secondary sources (e.g. news reports, social media). Such maps are useful for model calibration, validation, and flood susceptibility assessment (Cohen et al., 2018; Munasinghe et al., 2018). The USFIMR web portal provides more information on each flood, the specific sensor, as well as supplementary data including NED elevation and upstream
NWIS hyperlinks (http://sdml.ua.edu/usfimr). A catalogue of the USFIMR maps used in this study can be found in the appendix Table 1. Here it is noted that the FloodID assigned to each flood in this analysis is consistent with those used in the USFIMR and not necessarily sequential. Using these datasets, we address three main research questions:

1. Is the HAND method viable for continental flood mapping?
       2. Are the current NWM forecasts, HAND layers, and SRCs adequate for providing operational flood guidance?
       3. How do these three datasets contribute to errors in simulation?

**3. Methods**

This section will describe the creation of our simulation maps and the metrics used to evaluate accuracy.

**3.1 Floodplain Maps**

For each USFIMR map, the flood polygon was projected from NAD83 / Conus Albers (CRS 5070) to a WGS84 coordinate reference system (CRS 4269). For each shapefile, a clipping extent, derived as a concave hull was created to ensure all pixels being evaluated were within the USFIMR classification
bounds. These clipping extents were then used to subset the NHD and extract a list of COMIDs and HUC6 identifiers.

For each HUC6, the HAND, catchmask, and rating curve products were downloaded from the UT server and processed. The timestamp of each USFIMR satellite image was used to query the needed
NWM v1.2 reanalysis values by COMID and generate an inundation map using the HAND methodology (section 2.1).

For each event, a waterbody mask was created by combining the perennial NHD water bodies (NHD Fcode 39004, 39009) and NHDAreas (NHD FCode 40300, 40307, 40308, 40309) in each extent. These features were then rasterized to the 10m HAND grid using the fasterize R package (Ross, 2018). All cells that were not within the concave hull or covered by a waterbody mask, were set to NA prior to comparison. This process was repeated for each USFIMR flood and the process is formalized in the AOI, HydroData, and Flood Mapping R package (Johnson, 2018, Johnson, 2019a, Johnson, 2019b).

## 3.2 Catchment scale

In this paper we are also interested in isolating the biases stemming NWM forecasts and the HAND/SRC layers using gaged catchments. To do this, we identified all USGS National Water Information System (NWIS) gages within the concave-hull of each USFIMR polygon. For each gage, the collocated COMID was identified using the Network Linked Data Index (HydroData binding). Gages were kept if there was a NWIS and NWM record for the date/hour of interest and the complete catchment was contained within the USFIMR concave hull. In total, 54 unique catchments were identified.

For each catchment two unique flood maps were produced. The first used the NWM forecast and the SRC rating curves, while the second used the NWIS observation. Similar, to the floodplain maps, a waterbody mask was used to remove known cases of standing water.

## 3.3 Metrics used for comparison

With our floodplain and catchment-level flood maps we are interested in quantifying two types of agreement. The first is agreement between an observed (USFIMR) and simulated map. The second is how maps change with changing inputs.

### 3.3.1 Comparing Simulations to Observed Events

To compare simulations to observed extents, a simulated and observed map are intersected and classified into the four categories seen in Table 1. In these categories, 'W' refers to a wet cell and 'D' refers to a dry cell. The first letter in each category refers to the state of the cell in the observed map while the second refers to the state of the cell in the simulation.

Table 1: Flood Comparison Confusion Matrix.

|  | Modeled Wet | Modeled Dry |
|---|---|---|
| Observed Wet | WW (true wet) | WD (false dry) |
| Observed Dry | DW (false wet) | DD (true dry) |

With this classification, four metrics were established to describe the area ratio, rate of accuracy, rate of overprediction and rate of underprediction. The last three of these are derivatives of the fit statistic used by Sangwan and Merwade (2015).

The spatially agnostic Area-Ratio (Eq. (2)) helps establish if the simulated extent is inundating the correct number of cells regardless of spatial accuracy. Any value less than one indicates the NWM-HAND method is not inundating enough cells while a value greater than 1 indicates too many cells were inundated.

$$AreaRatio = \frac{Total\ Wet\ Simulation\ Cells}{Total\ Wet\ Observed\ Cells}$$   (2)

Overall Accuracy is quantified as the number of cells that were correctly identified as wet (WW) divided by the number of all wet cells in the simulated and observed rasters (Eq. (3)):

$$Accurate\ (A) = \frac{WW}{WW+WD+DW}$$   (3)

Similarly, areas of Underprediction are quantified as:

$$Under\ (U) = \frac{WD}{WW+WD+DW}$$   (4)

And areas of Overprediction as:

$$Over\ (O) = \frac{DW}{WW+WD+DW}$$   (5)

Together the Accurate, Over and Underpredict metrics form our AOU statistics. The use of a common denominator across these statistics assures the summation of AOU equals 1. The choice of a 2D fitness statistic (examining only the extent of flood, as opposed to depth and timing of flood propagation) is
governed by the aerial imagery products available (which only captures the extent of the flood, at a singular point in time). By electing this form of evaluation, we only analyze the strengths of NWM-HAND simulations at the given time-step coinciding with the time of image capture (not necessarily peak flooding).

**3.3.1 Comparing Simulation to Simulation**

In our 54 gaged catchments we have simulations driven by NWM forecasts, simulations driven by NWIS observations, and observed USFIMR events. To compare how the inclusion of known streamflow values change the overall accuracy of the simulated map, we calculate the AOU metrics for each set and then compute the standardized difference in the accuracy (A).

$$\Delta A = \frac{A_{NWIS} - A_{NWM}}{A_{NWIS}} \tag{6}$$

Positive values indicate the use of NWIS data increased the accuracy of the produced flood map while negative values indicate the use of NWIS decreased overall accuracy. Finally, a normalized mean error (nME) and normalized mean absolute mean error (nMAE) will be used to compare any two
measurements (e.g. stage or streamflow). In these, the control is the observed value and simulation is the estimated value.

$$nME = \frac{Control - Simulation}{Control} \tag{7}$$

$$nMAE = \frac{abs(Control - Simulation)}{Control} \tag{8}$$

## 4. Results

### 4.1 Floodplain analysis

First, we evaluate the ability of the NWM-HAND method to produce the correct amount of inundated area at the floodplain scale by plotting the area ratios (Eq. 2).

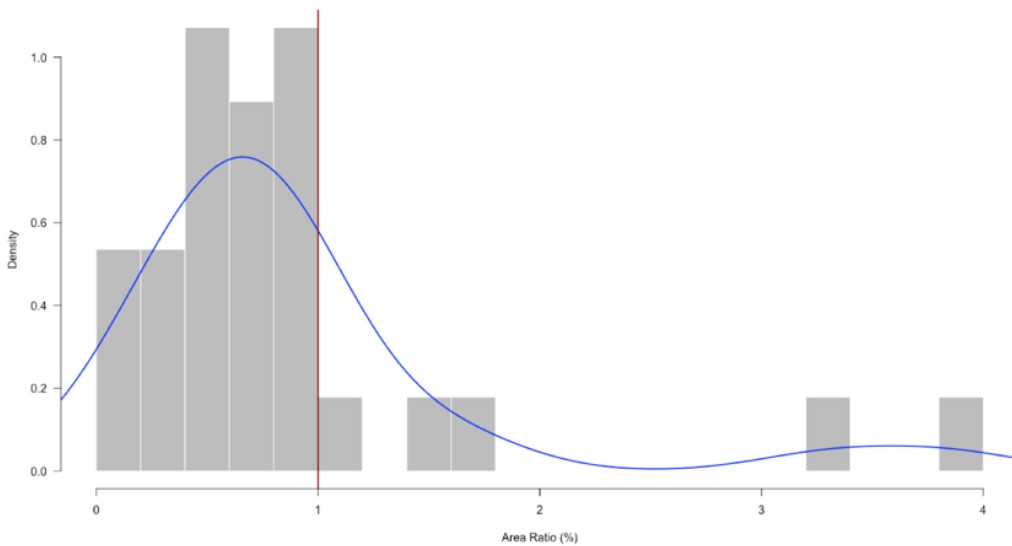

*Figure 2: A histogram of area-ratios for 28 floodplains show a tendency for the NWM-HAND to underpredict flood extents.*

Figure 2 shows that the overall median of our sample was 73% indicating a strong bias towards underprediction. Moreover, 82% of the floodplain maps had an area-ratio less than one. While there are fewer cases of over prediction (n = 5), the mean area-ratio of 230.4% indicates that while instances of overprediction are less frequent (~18% of our cases), errors are generally much larger.

From a spatially explicit perspective we evaluated how the NWM-HAND approach captures cell-level inundation using the AOU metrics. Table 2 shows that, of all flooded cells (in either the simulation or observed data), an average of 53% are underpredicted, 27% are overpredicted 19% were correctly identified.

*Table 2: Mean and median AOU statistics for the 28 floodplain studies.*

|  | **Median** | **Mean** |
|---|---|---|
| **Accurate** | 0.15 | 0.19 |
| **Over** | 0.25 | 0.27 |
| **Under** | 0.47 | 0.53 |

Figure 3 shows the AOU metrics for each floodplain as a stacked bar-plot further emphasizing the tendency toward underprediction across the entire floodplains.

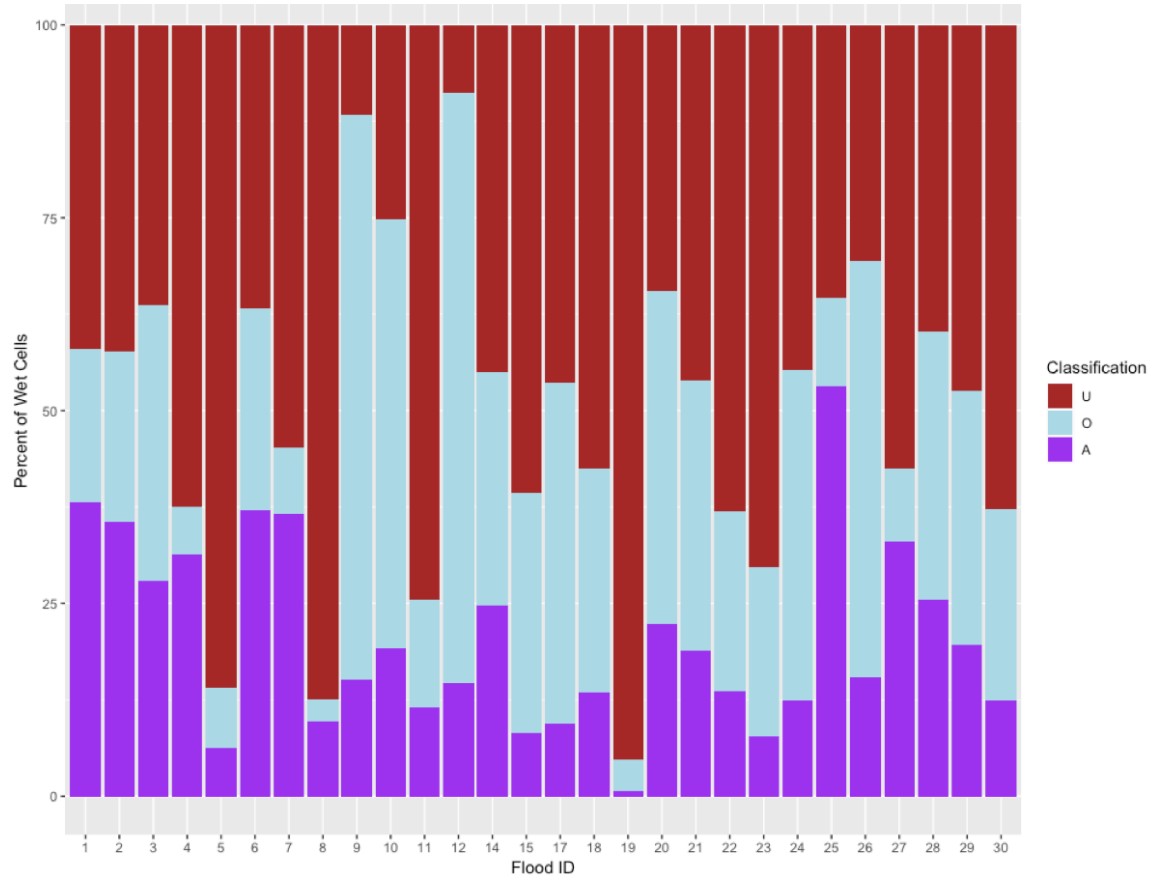

 *Figure 3: Stacked bar plot of the Accuracy, Over, and Under predict (AOU) metrics for each floodplain case study. Each bar represents 100% of the inundated cells in the intersected simulation and observed events.*

These statistics show a clear signal that the NWM-HAND method is limited in its ability to capture inundation and is almost twice as likely to underpredict than overpredict the state of a missed cell. Noting the stated intention of NWM-HAND as an operational flood mapping framework for guidance, 270 this tendency raises concerns.

**4.2 Catchment Area Comparison**

In this section we analyze the 54 gaged catchments to better understand the influences driving performance. As a reminder, for each of these catchments we have two simulated extents, one using the NWM forecasts and one using the NWIS observed flows. First, we evaluate the ability of the NWM-275 HAND method to produce the correct amount of inundated area by plotting the area-ratios comparing the total number of wet cells in the simulated and observed extents (Figure 4).

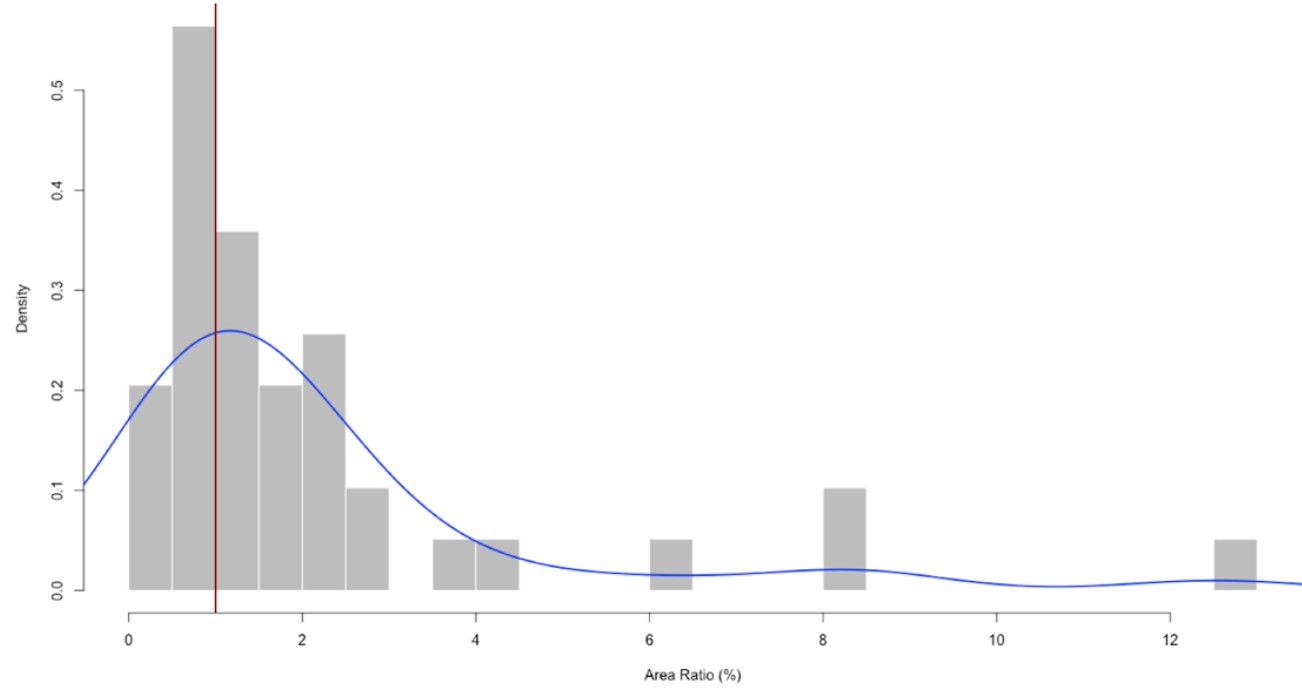

*Figure 4: A histogram of area-ratios for 54 catchments show a tendency for the NWM to slightly overpredict flood area and a long right tail of extreme over prediction.*

When compared to Figure 2, we see the peak in the density curve now sits much closer to 1 with a more equal proportions on either side. This indicates that at the catchment level, NWM-HAND is better able to capture the total area of inundation. However, we see that the over predicted cases are more extreme causing the right tail to elongate.

### 4.2.1    Influences of NWM streamflow inputs

The median and mean statistics for the NWM and NWIS-driven AOU metrics are presented in Table 3.

*Table 3: Mean and median values for NWM and NWIS AOU metrics and R² values for linear fits between NWIS and NWM..*

|  | **Median** NWM / NWIS | **Mean** NWM / NWIS | **R2** NWIS~NWM |
|---|---|---|---|
| **Accurate** | 16% /16% | 25% / 25% | .93 |
| **Over** | 45% /47% | 44% / 47% | .82 |
| **Under** | 26% / 21% | 31% / 28% | .75 |

Table 3 highlights that using known streamflow values generates no change in average accuracy with a slight increase in overprediction and corresponding decrease in underprediction (~3%).  To better

understand how well the AOU metrics of the NWIS-driven maps explain the variation in the NWM-driven maps, linear models were fit to describe ($NWIS_A \sim NWM_A$), ($NWIS_O \sim NWM_O$) and ($NWIS_U \sim NWM_U$). The $R^2$ values of these models can be seen in Table 3 as well. From these results we can conclude that approximately 7% of the variation in accurate prediction can be attributed to inaccuracies in the NWM forecasts. Similarly, 18% of the variation in over prediction and 25% of the variation in underprediction can be explained by inaccuracies in the NWM. All said, this suggests that even with perfect streamflow inputs, current implementations of HAND and SRCs are only capturing ~25% of the inundated cells correctly, and a minimal percentage of can be explained by variation in NWM forecasts. Moreover, the errors in NWM forecasts have a limited influence on if cells are accurately captured but a strong influence on whether missed cells are under or over predicted.

Table 3 also reveals that, regardless of streamflow input, the tendency at the catchment level is to over predict (47%) rather than under predict (28%) inundation. This is a opposite pattern seen when looking at the entire floodplain maps despite the fact that the average tendency of the NWM is to under predict streamflow in these events (nME of -20%).

To confirm that our conclusion that NWM forecasts have a minimal impact of inundation accuracy, we need to ensure that the adverse impacts of bad simulations are not masked by many accurate forecasts. To better understand the relative bias introduced by the NWM we grouped our 54 catchments by their nMAE. For each set of basins grouped by nMAE, the number of gages and absolute mean change in accuracy ($\Delta A$; Eq. (6)) are shown in Table 4.

*Table 4: Absolute mean $\Delta A$ gage count and grouped by nMAE between NWIS and NWM streamflow.*

| nMAE range | Gages (cumulative %) | Absolute mean $\Delta A$ % |
|---|---|---|
| 0 - 10% | 9 (16.67%) | 0.68 |
| 10 - 20% | 11 (37%) | 1.91 |
| 20 - 30% | 4 (44%) | 4.18 |
| 30 - 40% | 3 (50%) | 6.36 |
| 40 - 50% | 3 (55.8%) | 6.81 |
| 50 - 60% | 6 (66.7%) | 8.39 |
| 60 - 70% | 8 (81.49%) | 31.77 |
| 70 - 80% | 1 (83.3%) | 69.52 |
| 80 - 90% | 3 (88.90%) | 30.15 |
| 90 - 100% | 2 (92.60%) | 84.36 |
| >100% | 4 (100%) | 391.42 |

Looking at the absolute mean change in accuracy between the NWIS and NWM maps, there is evidence that the influence of the NWM on NWM-HAND accuracies are minimal until nMAE exceeds 60% when there is a clear jump in the absolute mean $\Delta A$ from less than a 10% relative loss to more than 30%.

When the NWM exceeds nMAE of 60%, predictable patterns of error occur. Figure 5A (Flood ID 4), shows a case where NWM-HAND under predicts inundation due to an underpredicted NWM forecast

while figure 5B (Flood ID 9) shows a case where an over predicted NWM forecast results in over predicted flood extents.

In addition to errors in streamflow magnitude, NWM timing errors can introduce issues related to flood inundation forecasting and its evolution. Even though we did not explicitly study the ability of NWM-HAND to capture the temporal evolution of flood extents in this study, a few instances where the NWM improperly estimated the receding limb of the hydrograph, showed the inability of NWM-HAND to capture ponded and receding waters leading to issues of under and over prediction respectively.

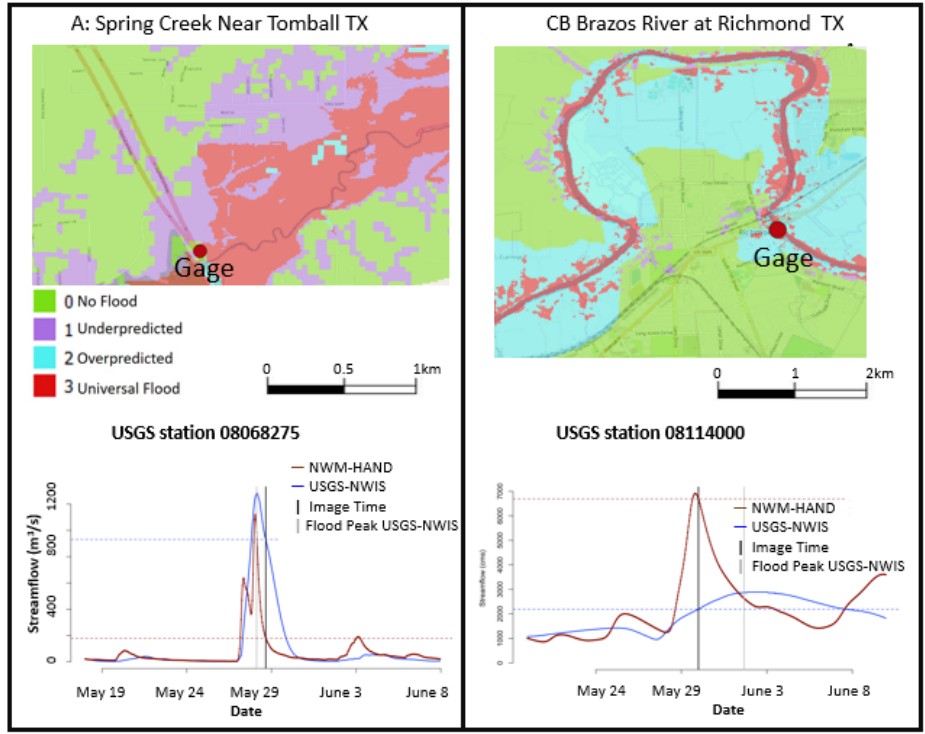

Figure 5: Prediction errors due to NWM flow forecast when nMAE exceed 60%: (A) NWM underprediction results in underpredicted flood extents (B) NWM overprediction results in overpredicted flood extent.

At this point we have found that at the floodplain level, NWM-HAND tends to accurately capture ~19% of inundated cells with almost twice the chance of under then over predicting a missed cell state. In contrast, at the catchment level NWM-HAND is able to accurately capture ~25% of the inundated cells with a more equal tendency to over and underpredict missed cell states, with a stronger tendency to over predict.

Dissecting this divergent pattern lead us to thinking about the general state of gaged reaches. Most often gages are put on larger rivers and less so on smaller reaches and tributaries. In fact, of the 54 catchments in our study, 90% were on a reach with a Strahler order of 4 or higher. In contrast, when all catchments

in each of our USFIMR floodplains were aggregated, low order reaches made up, on average, 81% of the total stream networks.  In many respects, this makes our catchments a biased sample favouring high order reaches.

Therefore, our hypothesis is that the difference in over and underprediction patterns seen between the floodplains and catchments can be explained by the difference in the sample and population. That is, low order reaches tend to underpredict while higher order reaches are more neutral with a tendency towards over prediction. From here on out, lower order reaches refer to those with a Strahler order of 3 or less, and higher order refers to those with a Strahler order or 4 or higher.

### 4.2.2 Underprediction in lower order reaches

A visual analysis of each of our floodplains reveals a distinct pattern of under prediction in lower order reaches, however due the general nature of gaged reached we only have one gaged catchment of a 2nd (Fig 6, Flood ID 8 USGS Station ID 8068325). At this reach, the NWM estimated 50 m³/s (SRC stage = 1.82 m) flowing through the channel. The recorded NWIS flow is 80 m³/s (nMAE = 0.38, SRC stage = 2.43 m). A cross section was made across the HAND raster at this reach indicating a stage of 3.3 m is required to inundate the right bank and 3.8 m is needed to inundate the left bank to the levels seen in the USFIMR.

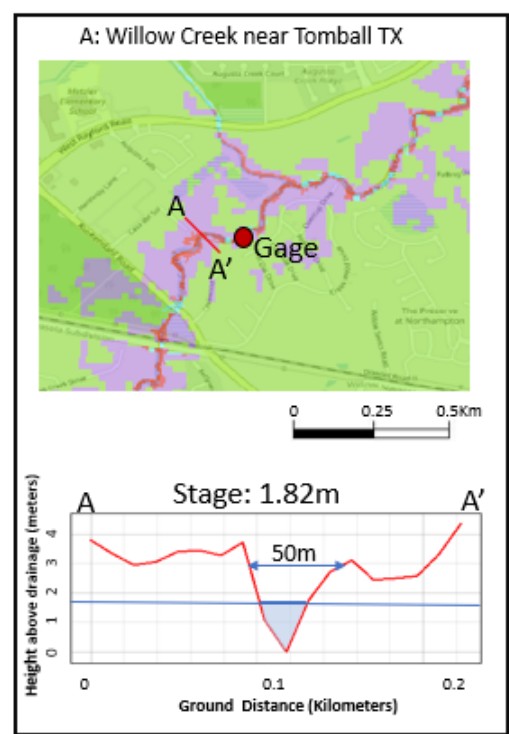

*Figure 6: Inaccurate synthetic rating curve relationship causes underprediction in a 2nd order stream*

In this case, even if the NWM had accurately predicted streamflow, water would have remained confined to the channel resulting in underprediction. Recognizing a mismatch between the known Q (80 m$^3$/s) and stage (3.8 m) we explored the assumptions driving the Manning's Equation SRC for this reach (Eq. (1)). Keeping slope (NHD attribute) and the cross-sectional area required to generate a stage of 3.8 m constant, we independently varied the roughness coefficient (N) and the hydraulic radius (via the wetted perimeter), solving for a Q of 80 m$^3$/s. In doing so we found that the SRC relationships are generally insensitive to changes in hydraulic radius (needed to be increased by a factor of 10), but were sensitive to changes in Manning's N. Specifically, we found that an N of 0.16 was needed to achieve a observed streamflow-stage relationship at this reach. Based on this representative example we suggest that systematically increasing roughness in lower order reaches may help mitigate underprediction. Other research that applies N values by stream order offer similar estimates to what was found in our example (N= 0.12 for second order reaches; Li, 2016). Certainly, more research is needed on the relative role of roughness in the NWM-HAND method and the best way to optimize it across different stream orders and geographies.

### 4.2.3 Overprediction in areas of low relief

The other reoccurring error we noticed in our analysis was the extreme right tail seen in the area-ratio plot for catchments (Figure 5). When we isolated the poor performing catchments it became evident that the extreme over prediction occurred in areas of low relief. Overall, we found two types of low-relief cross-sections: those that are hyper-sensitive to errors in stage, and, those that are fundamentally limited by the terrain representation in the HAND layer.

### 4.2.3.1 Sensitivity to SRCs

Since HAND is not a physical model, it is unable to conserve volume through space or time. In areas of low relief, where many cells have similar if not equal HAND values, small errors in stage can have disproportionate errors in inundation extent at the 10m grid cell resolution. The Washita river at Anadarko OK (Figure 7A) is a 6th order stream with relatively uniform overbank terrain. The NWM estimates 473 m$^3$/s (SRC stage = 3.81 m) while the recorded flow was 495 m$^3$/s (SRC stage = 3.88 m). The nMAE of 4% suggests the source of overprediction originates in the terrain layer or SRC and not the NWM forecast. For this reach, we optimized the SRC for the given cross-sectional area at an observed stage of 1.2 m and solved for a Q of 495 m$^3$/s. To achieve this, N was reduced to 0.005, effectively increasing the carrying capacity of the river. This solution supports the pattern of decreasing N in higher order rivers and this emphasizes that fact that in areas with low relief, small errors in stage and or discharge can result in extreme overpredictions.

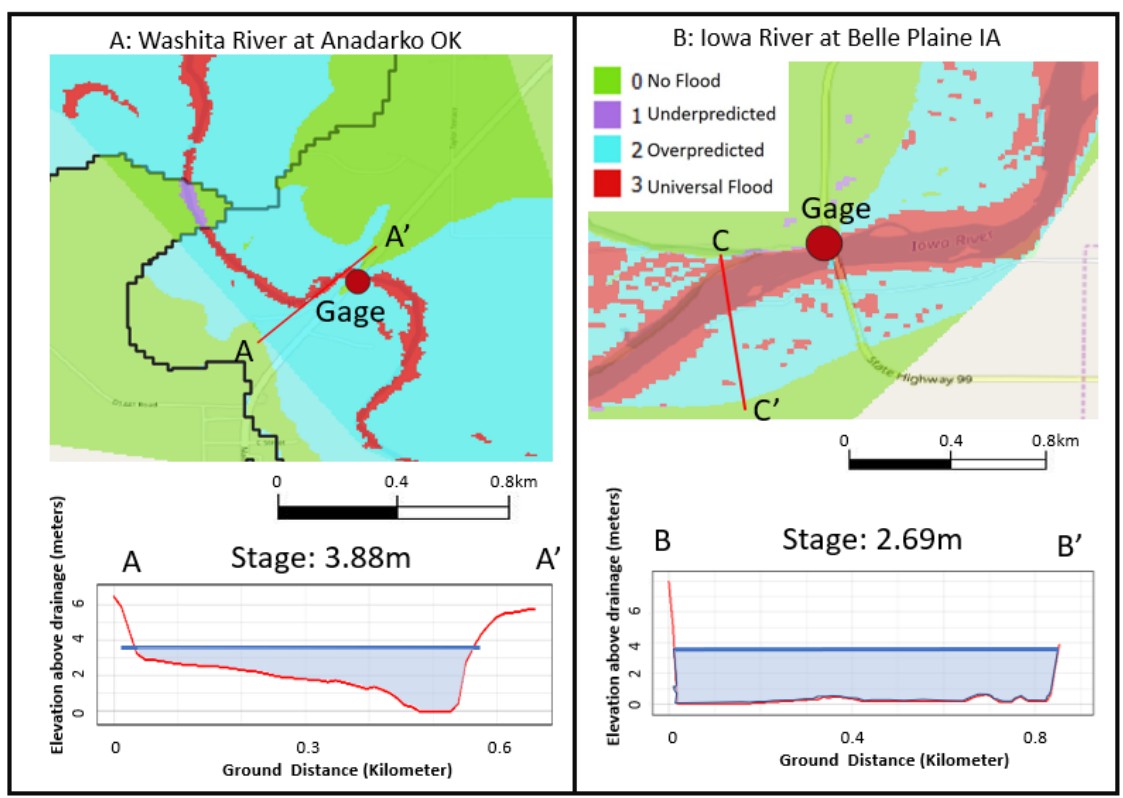

Figure 7: Over prediction in catchments with low relief. (A) This catchment is influenced by a hyper-sensitive synthetic rating curve relationship (B) This catchment exemplifies a case where the HAND layers will be unable to accurately capture flooding extents due to essentially 0 relief.

### 4.2.3.2 Concerns with raster resolution in areas of low relief

We examined a second type of low relief profile where optimizing the hydraulic parameters has no effect due to the form of the HAND raster. This is illustrated with a cross section along the Iowa River at Belle Plain IA (Figure 7B Flood ID 17). The NWIS observed streamflow for the reach was 971 m$^3$/s (stage = 2.69 m) while the NWM predicted streamflow was 958 m$^3$/s (stage = 2.66). The issue with this reach (and others like it) is that the area currently simulated as flooded, will flood for any stage exceeding 0.5 m regardless of the volume of water within the channel. It is in these regions that the 10 m grid cell resolution of the underlaying DEM and absolute vertical accuracy of the NED (2.44 meters; FEMA, 2007) appear most limiting.

We suggest two possible improvements for low relief catchments. The first is refactoring of the NHD catchments (or the effective HAND catchments) into more uniform units avoiding large, flat drainages and instances of many small units. Tools for NHD refactoring are already in development (Blodgett, 2018), and the idea of refactoring speaks to issues to be raised in section 4.6. A second possible

alternative to refactoring is to make use of the NWM velocity and flow estimates to define cross sectional areas from the NWM forecast:

$$Cross\ Sectional\ Area = \frac{Q}{Velocity} \qquad (9)$$

The intention would be to allow the physical model (NWM) and routing-routines (WRF-Hydro) to deal with issues of volume preservation. The resulting cross-sectional areas could be used as an Area-Stage rather than Q-Stage look up within the existing SRCs. This would work around some of the issues with
roughness (outsourcing to the NWM) while capitalizing on the observed accuracies in the floodplain cross sections. Moreover, by controlling for the volume of water in the channel instead of the height, low lying areas will be less prone to exaggeration. Such a change would require an understanding of how the NWM is handling hydraulics and thus velocity and a test of how variations in velocity impact volume estimation. Both are interesting pursuits in their own right but out of scope for this paper.

## 4.3 Other source of error/influence

While the primary purpose of this study was the impacts of NWM and HAND data layers and assumptions on NWM-HAND flood simulations, this sections looks at two other issues that influence flood simulation and our accuracy metrics: namely, the data models underpinning the HAND layers and
the limitations in remotely sensed ground truths.

### 4.3.1 Data Models: Use, Limitations, and Adaptions

The Digital Elevation Model (DEM) used to prepare the current HAND layers is a 10 m product that effectively smooths the surface of the earth. In areas of the low relief, higher-resolution elevation data would be able to produce more sensitive rating curves. In other areas, attention to the methods and
datasets (streamlines) used to condition the DEM should be considered.

With respect to streamlines, it is important to recognize that the NHD was developed as a cartographic representation of the nation's waterways and using a cartographic toolset for hydrologic modelling and routing applications has inherent limitations. In Figure 8A (Flood ID 12), the NHD does not capture the
meandering segment of the Canadian River's main channel. In such a situation, even with accurate NWM forecasts and rating curve relationships, the predicted flood extent will be spatially misaligned. Similar situations can also occur in areas with braided channels or wide estuaries. Standard DEM-based stream delineation tools (like those used in NHD) only allow one channel to be designated for each reach and often result in a poor representation of network density or/and channel width. Prediction
errors resulting from network density/missing reach issues can be seen in FloodID 8, while errors due to channel and or coastal feature representation can be seen in FloodIDs 19, 23 and 24.

On a more theoretical side, the use of a cartographic datasets as model base layers often involves some form of manipulation. Up until this point, the NHD has been used in its original form without

considerations for the implications on modelling. As the NHD becomes a more central component of the NWM framework, the goal of creating more specific and consistent modelling units while preserving the network is of value. To illustrate this, Figure 8B shows a collection of catchments in FloodID 17. The two principle catchments experiencing flooding in this image are highlighted in red. Both catchments are long and narrow making them inconsistent with the others in the region. One issue
with these long catchments can be seen in the leftmost red catchment. To the north of the catchment (in the 'black' catchment) we see a large swath of under predicted flooding (purple). Looking at the NHD reach draining the black catchment it is easy to see that the area of missed flooding did not stem from the catchment reach but from the long red catchment. This implies that flood water crossed the catchment boundaries in a way that cannot be captured by HAND. Ultimately, integrating cartographic,
sensor, and model domain data will be an issue for all involved fields in the near future, and in some respects lay out the challenges for the next 20 years (Clarke et al, 2019).

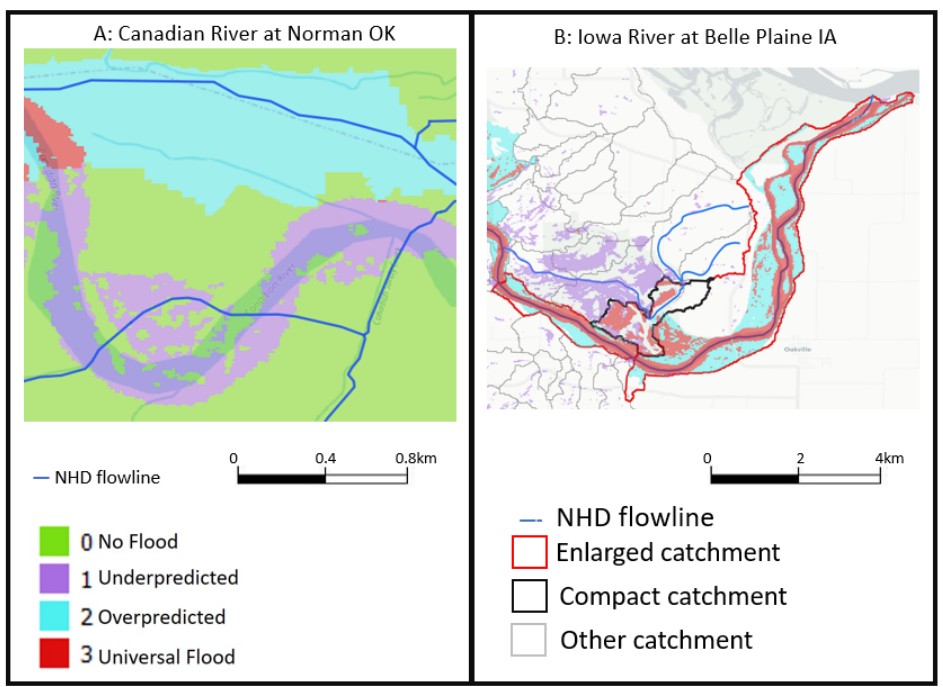

*Figure 8: Unaccounted for challenges when using the NHD as a modelling fabric. (A) this catchment shows a case where the NHD/ NED is not capturing the true meander of a river channel (B) This set of catchments highlights the influence of long non-uniform catchments and challenges with catchment definition.*

### 4.3.2 Remote sensing (RS) Challenges

In all analyses we treated the USFIMR maps as ground truth. For the sake of a robust comparison it is worth highlighting some of the issues that arise in the creation of remotely sensed inundation maps and
their consequences in these comparisons.

### 4.3.2.1 Pixel misclassification

The spectral similarity of dark vegetation and water pixels in optical satellite imagery contributes to uncertainties in both directions. Most often, these errors resulted in isolated patches of dark vegetation being classified as water (Figure 9A FloodID 2). Other cases of pixel misclassification occurred in FloodIDs 1, 3, 16, 17, 18 and 23. Water was also incorrectly classified in a few notable cases. For example, pixels clearly within the main channel of the Mississippi river were classified as unflooded in Figure 9B (Flood ID 13). In this example, the NWM-HAND methodology correctly identified such locations as being flooded (despite over predicting the overall flood extent). These error types were also present in floods 19 and 23.

### 4.3.2.2 Image artefacts

Image artefacts also impact our comparison. The first is the well-known scan line error present in LANDSAT 7 images processed after 2003 (Scaramuzza and Barsi, 2005). Despite gap filling, some data gaps still remain. The gaps created by the scan line error (Figure 9C; Flood ID 4), created situations where both methods missed a flood or HAND-NWM alone predicts a flood, even though flooding could be detected from surrounding pixels. The overall extent covered by scan line gaps did not have a large impact as this issue affected only a relatively small percentage of the area in one case study.

The second artefact encountered was cloud cover which had a strong effect in one case study (Figure 9D; Flood ID 5). The image used to generate the USFIMR map was taken from a LANDSAT scene with 40% cloud cover, making it difficult to accurately classify some segments of the image. This comparison also contained large regions of freshly melted snow (visible in the LANDSAT image as well as from the road network and farm fields in the classified image). Cloud related classification errors also affected FloodID 15, but to a lesser degree.

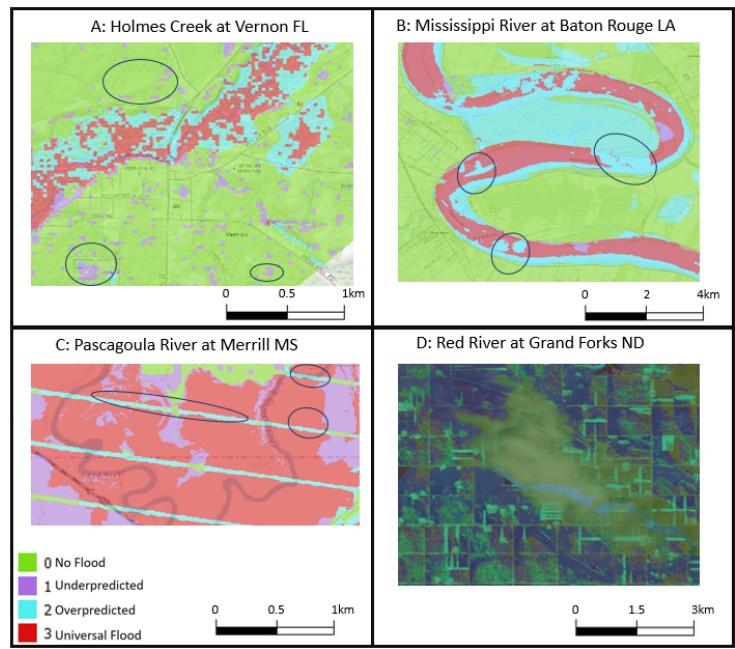

*Fig 9: Errors due to classification and image artefacts: (A) Dark vegetation misclassified as water due to spectral similarity. (B) Misclassified channel segments due to spectral similarity. (C) Scan line errors in input RS image create gaps in flooded extent. (D) clouds and melted snow create ambiguities in flooded extent recognition.*

In all cases, these errors only occurred in minor portions of a few floods used in the analysis. We raise them not because they adversely impacted our results but for transparency and to provide readers with an understanding that no form of ground-truthing is infallible. The rigorous development, successful use in other studies, and our careful evaluation of each flood extent give us confidence in their overall accuracy. More importantly, such a resource is the only option for validating modelled inundation extents at the scales we are exploring. As the NWM-HAND method continues to improve, be developed, and further validated, resources like the USFIMR will only become more necessary.

## 5. Discussion

In its current form NWM-HAND is capturing 19-25% of inundated cells accurately. At the floodplain resolution, NWM-HAND has a larger tendency to under (53%) then over predict (27%). In contrast, we saw a larger tendency to overpredict (47%) then underpredict (28%) inundation at the catchment level. We argue this tendency to underpredict area at the floodplain resolution was an aggregate result of consistently underpredicting many, small lower order catchments.

While the overall NWM-HAND methods certainly have a long way to go, we contend the results demonstrate the NWM-HAND approach is useful for general guidance and risk identification but may not be suitable for pixel level analysis, resource allocation, or risk-communication. Despite the limited accuracy found in this study, the NWM-HAND is quite an achievement that should not be discounted.

The fact that an uncalibrated continental-scale model can rapidly generate flood inundation maps is of great value and more importantly, the base layers and conceptual framework underpinning the model offer the research community a resource to improve, modify, and manipulate.

Looking towards a future of research and development and the inevitable next version of the base layers, issues of availability arise. Currently, accessing and combining the needed NWM-HAND products is cumbersome, and for regions straddling one or more HUC6 can be data and processing intensive. While services like the Flood Mapping R package can streamline some of the steps, disseminating data by HUC6 is not a convenient choice for users. Instead HAND products could be distributed via a web service like those used to distribute climate data and NWM gridded output - or - built into services like the CUAHSI sub setter (CUASHI, 2019). With workable foundations now in place, engaging with the communities that will use these products for research, map creation, and ultimately in the field seems like a next step worth considering.

Returning to our research questions, we believe the HAND method is viable for continental flood mapping aimed at guidance, and not cell by cell decision-making. With this caveat in mind, current NWM forecasts are accurate enough in ~2/3 of our situations (a condition that will arguably improve in the operational model with data assimilation and stronger calibration) and the HAND terrain layers are adequate in all but areas of low relief. The synthetic rating curves are the source of most error, but can be greatly improved through a closer examination of Manning's N. In all of our tests, the existing cross-sectional geometries could be altered to provide correct Q when varying Manning's N within a range of 0.0001 and 0.2. To us, opportunity and potential of the existing data and methods should be weighted more heavily in the mind of readers then the reported low accuracies.

## 6. Conclusion

This study offers a high-level evaluation of the confidence we can place in the operational NWM-HAND forecasts. In its current state the NWM-HAND methods have limited ability to accurately capture inundation and its skill is more constrained by the terrain and SRC inputs then NWM accuracy.

At floodplain level, NWM-HAND tends to underestimate flooding area (82% of case studies). Of these, the mean likelihood of underpredicting a missed cell (53%) is twice that of overprediction (27%). At the catchment scale, NWM-HAND was better able to capture the total area of inundation but was more likely to overpredict (44%) than underpredict (31%) missed cells. We attribute this disparity to systemic underprediction in lower order reaches. An analysis of NWM-HANDs sensitivity to changes in Manning's N and cross-sectional geometries indicate that SRCs are insensitive to changes in hydraulic radius (ergo wetted perimeters) but are very sensitive to changes in Manning's N.

As a general rule we believe the default Manning's coefficient used by HAND are too small in low order reaches and are arguably too large in higher order reaches. While we only tested this on the small sample of available catchments, the theory behind the Manning values, and the consistent propensity to

under predict low order reaches would support such a change. In all cross-sectional geometries we tested, observed streamflow / stage (NWIS / USFIMR) relationships were achievable with a variable N, save those with zero relief.

We further investigated the level of bias in NWM-HAND maps coming from the NWM forecasts and found the method to be relatively insensitive until the normalized absolute error between NWM and observed flow exceeded 60%. Above this threshold, predictable patterns of large under or over prediction occur in the direction of the NWM miss.

Another reoccurring error in our simulations was over prediction in catchments with low relief. Some of these catchments allow incremental changes in stage to have large impacts on inundation extent. We suggest that a refactoring of the NHD to smaller more uniform units, and a consideration for how volume rather than stage may be used to fill catchments, may be possible solutions. Moreover, the former offers some solutions to issues arising from treating the cartographic NHD as a modelling infrastructure. That said, neither of these are trivial to accomplish or validate.

While the current ability to capture accurate inundation is limited, our analysis revealed no fundamental errors that would render NWM-HAND unusable. Instead, we found that with modifications, accurate inundation is generally achievable at the level needed for guidance and emergency management. Therefore, we end with great optimism that further development, research and evaluation around the NWM-HAND method can deliver on the promise of a CONUS scale flood modelling enterprise.

## 7. Code Availability

The AOI, HydroData, and Flood Mapping are all available at their respective GitHub repositories at (https://github.com/mikejohnson51/).

## 8. Data Availability

USFIMR data is available at (http://sdml.ua.edu/usfimr). NWM reanalysis products are available at (https://registry.opendata.aws/nwm-archive/). Processed HAND products are available at (https://web.corral.tacc.utexas.edu/nfiedata/).

## 9. Appendix

Table 1. Flood events used for comparison

| FloodID | River name | Flood date | Satellite sensor | State | Flood area (km$^2$) |
|---|---|---|---|---|---|
| 1 | Choctawhatchee River | 4-Jan-16 | Landsat 8 OLI | FL | 1493 |
| 2 | Holmes Creek | 4-Jan-16 | Landsat 8 OLI | FL | 129 |
| 3 | Pea River | 4-Jan-16 | Landsat 8 OLI | AL | 407 |

| | | | | | |
|---|---|---|---|---|---|
| 4 | Pascagoula River | 17-Mar-11 | Landsat 7 ETM+ | MS | 1692 |
| 5 | Red River | 18-Apr-97 | Landsat 5 TM | ND | 7777 |
| 6 | Brazos River | 28-May-16 | Landsat 8 OLI | TX | 74 |
| 7 | Spring Creek | 28-May-16 | Landsat 8 OLI | TX | 160 |
| 8 | Willow Creek | 28-May-16 | Landsat 8 OLI | TX | 48 |
| 9 | Washita River | 29-May-15 | Sentinel-1 VH Polarization | OK | 623 |
| 10 | Brazos River | 30-May-16 | Sentinel-1 HV Polarization | TX | 951 |
| 11 | San Jacinto River | 30-May-16 | Sentinel-1 HV Polarization | TX | 212 |
| 12 | Canadian River | 17-Jun-15 | Sentinel-1 VH Polarization | OK | 739 |
| 14 | Trempealeau River | 26-Sep-10 | Landsat 5 TM | WI | 276 |
| 15 | Wisconsin River | 26-Sep-10 | Landsat 5 TM | WI | 8350 |
| 17 | Iowa River | 26-Sep-16 | Landsat 8 OLI | IA | 3541 |
| 18 | Maquoketa River | 26-Sep-16 | Landsat 8 OLI | IA | 2249 |
| 19 | Mississippi River | 26-Sep-16 | Landsat 8 OLI | IA | 1937 |
| 20 | Wapsipinicon River | 26-Sep-16 | Landsat 8 OLI | IA | 4148 |
| 21 | Minnesota River | 1-Oct-10 | Landsat 5 TM | MN | 7876 |
| 22 | Redwood River | 1-Oct-10 | Landsat 5 TM | MN | 1414 |
| 23 | Mississippi River | 3-Oct-10 | Landsat 5 TM | MN | 6957 |
| 24 | Ashley River | 13-Oct-16 | Landsat 8 OLI | SC | 1804 |
| 25 | Black River | 13-Oct-16 | Landsat 8 OLI | SC | 431 |
| 26 | Cooper River | 13-Oct-16 | Landsat 8 OLI | SC | 2096 |
| 27 | Lumber River | 13-Oct-16 | Landsat 8 OLI | SC, NC | 1116 |
| 28 | Neuse River | 15-Oct-16 | Landsat 8 OLI | NC | 1054 |
| 29 | Tar River | 15-Oct-16 | Landsat 8 OLI | NC | 1702 |
| 30 | Rio Grande River | 14-Jul-10 | Landsat 7 ETM+ | TX | 6836 |

## 10. Competing Interests

The authors declare that they have no conflict of interest.

## 11. Acknowledgements

We would like to thank David Tarboton and David Blodgett for their constructive reviews that helped shape this paper. USFIMR is funded through the NOAA National Water Center (NWC) via the UCAR COMET Program (2016/17), AOI, HydroData, and Flood Mapping R packages were developed through the NOAA National Water Center (NWC) and UCAR COMET Program (2017/18).

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
