# Peer review of "An Integrated Evaluation of the National Water Model (NWM) Height Above Nearest Drainage (HAND) Flood Mapping Methodology"

_Natural Hazards and Earth System Sciences, 2019_

## Referee Comment (RC1) · David G. Tarboton (Referee) · 18 May 2019

**General comments**

This paper evaluates the National Water Model (NWM) Height above Nearest Drainage (HAND) flood mapping methodology by comparing modeled inundation area with mapped inundation derived from satellite products for 30 flood events. This is a worthwhile comparison, as, given the importance of the NWM and its potential use for flood

warning, emergency response, and planning, it is important to know the uncertainty associated with its predictions. The comparisons presented in this paper are helpful in this regard, though there are limitations to the analysis, that merit caution in not over-interpreting the results. While the analysis is helpful, I feel that its claims of being "comprehensive" in the title and "first detailed" in the abstract are not justified.

The first caution is that the statistic (AFI, equation 1) used to quantify accuracy is generous and can introduce bias and arbitrariness. It scores the match between model and observed as the fraction of area modeled and observed to be flooded, plus the fraction of area modeled and observed to not be flooded. This weights area not flooded heavily in the calculation. Bias and arbitrariness are introduced as the non-flooded area depends on the extent of the convex hull used for the area analyzed. If the geometry is such that a large part of non-flooded area is included in the convex hull, then AFI will be inflated. I am concerned that the conclusion that the method can be used "quite confidently" is biased because of this, and the ranking of how well the different floods are modeled is uncertain due to the arbitrariness of the convex hull. See specific comments below for expanded discussion of this point.

Additional cautions are that errors can be due to errors in the NWM modeled discharge or errors in the HAND methodology. The paper mentions these and presents a table where the dominant cause is stated for some cases with poor AFI. However, the analysis does not separate the effect of the different errors, except anecdotally for some of the cases. Where there are discharge differences (such as illustrated in Fig 5 and 6) it should be possible to model inundation using the observed discharge and thus interpret which part of the error is due to discharge error, and which part of the error is due to HAND methodology error. I think that such separation is important in assessing limitations of the HAND methodology.

It would also be helpful, if in pointing out error (e.g. for small streams or flat areas) for the authors to offer ideas or suggestions towards correcting the problems. The detailed

comments below offer a few suggestions.

Given these general concerns, I find myself taking with caution the conclusion that NWM-HAND can be used "quite confidently" (Page 5, line 23, or P5 L23) or "general level of agreement" (P9 L28). Don't get me wrong. I do not want to dismiss this study. I believe that the HAND approach has considerable merit and that comparison studies such as this paper are important, but the analysis needs to be presented in a more complete, balanced and objective way to document where the approach is effective and to support the conclusions. I think that changes to address the arbitrariness problems of the statistic and to separate errors due to discharge versus HAND methodology, should be made before this manuscript is finally published. In the reviewer form, I characterized these as "major" revisions, but they may actually fall between major and minor and could, I think, be done fairly quickly.

**Specific Comments**

I do not think "Comprehensive" in the title is justified. 30 flood events are certainly a worthwhile study, but the study really only ended up reporting errors and did not dig into the causes in a way that merits use of the term comprehensive. E.g. the separation of problems due to discharge errors vs HAND errors was limited.

Error statistic. The authors acknowledge in part that the weighting of non-flood area in AFI is a limitation (discussion P9 L20-24) but did not do anything about it, claiming it is suitable for the purposes of this paper. I tend to disagree. For example, suppose that 10% of the area is observed to be flooded (Figure 2 suggests this to be about right), 90% of the area is not observed to be flooded. Say that the model predicts 10% of the area to be flooded, but mostly in the wrong location. Perhaps 1% of the area is observed and modeled to be flooded, with 9% of the area modeled to be flooded but not observed, and 9% of the area observed to be flooded but not modeled. This would seem like poor prediction, only getting 10% of the flooding in the right place. But with

these numbers 81% of the area that is not flooded is modeled to be not flooded, so the AFI score is 82% (81% non-flooded match + 1% flooded match) which would be interpreted as good. Another part of this problem is that the results are sensitive to the area of the convex hull, because that dictates the amount of non-flooded area in any evaluation of AFI. If the geometry is such that a large part of non-flooded area is included in the convex hull, then AFI will be inflated. For example, if given the numbers above the convex hull had been smaller so that 20% of the area was flooded, 20% modeled as flooded and 2% observed and modeled as flooded. This could be exactly the same comparison as above but removing half of the overall area from the non-flooded part of the convex hull. With these numbers the AFI score would be 64% (62% non-flooded match + 2% flooded match) falling towards the lower end of moderate, rather than good. I think there is a need to consider the potential distortion of the results and their interpretations due to this effect. This could be done by reporting the matching and non-matching areas for each case, as well as the total areas. Further, it may be better to use a statistic that does not consider non flooded area such as the fit statistic from Sangwan and Merwade (2015).

$$Fit(\%) = 100 \times \frac{F_{pred} \cap F_{obs}}{F_{pred} \cup F_{obs}}$$

Separate errors due to discharge errors from errors due to HAND methodology. Where observed discharge is available it should be used to generate a flood inundation map and differences between this and observed flooding examined as they are due to the HAND methodology alone. The paper states as its goal (P4 L17) "exploration of how errors in the methodology and its individual components may compound and propagate...". Nowhere does the paper separate out individual error components. Errors reported always combine effects of discharge error and HAND methodology error. This is an important shortcoming that should be addressed. Within the 30 study cases, which are due to poor discharge predictions, not a problem of the HAND methodology per se, but of the discharge forecasts. For example, in Fig 5, and the text (P6 L9) the

paper indicates that NWM simulated discharge was 50 m3/s, while the gage recorded 80 m3/s. This is a 60% difference and one should not expect a flood inundation mapping model to do well given such differences. An inundation map produced with the observed discharge would examine this. The paper could correct this by evaluating NWM discharges where there are streamflow gages. Section 4.2 suggests that some of this was done (P7 L20 "all NWIS stations within each flooded region ..."), but where are the results? The paper only presents a few discharge examples.

I do not think that Figure 5A and section 4.1.1 make a compelling case that raster resolution systematically suppresses stage in lower order reaches. Figure 4 is a bit of an oversimplification. The authors acknowledge this. While it is OK to make the point, it is not really consistent with Fig 5A. The observed inundation depth is stated to be 3.96 m (It would be good to state where this came from). Placing a stage of 3.96 m on cross section A-A' the river is notably over the banks and about 200 m wide. The cross section is thus not that poorly generalized by a 10 m DEM and a narrow channel such as suggested in Fig 4A, 5 m deep, seems inconsistent with the cross section for the observed stage and section A-A'. I would look to some other problem for this case. Discharge is part of it. But also Manning's n, slope, and general representation of the hydraulics involved (uniform flow assumption) may be problematic and warrant investigation. An additional point with respect to low order reaches, is that, being smaller, with smaller contributing areas, they likely produce less damaging floods.

Overprediction in areas of low-relief. Fig 5B. It would be helpful to diagnose what has gone wrong here. Was the large area mapped as inundated part of the NHD catchment used to compute the rating curve? A stage of 9.75 m is huge, and evidently an overestimate. Some analysis of why would be helpful rather than just generally saying this can be a problem. I think that overlaying NHD catchments used may be helpful, to see the areas used in calculating SRCs and whether the flood is extending

across catchments. Problems with roughness (Manning's n), slope, and the synthetic rating curve are all potential causes. For Fig 5C, this is a case where high resolution NHD streams may offer an improvement.

Data representation (section 4.1.3). When "NHD" is stated (P2 L27) I think that it is important to specifically indicate that it is the medium resolution NHDPlus dataset (I think that is what was used). There are high resolution NHD products becoming available that may improve matters, though there is work to be done to manage the scale of catchments used. NHDPlus medium resolution is nominally 1:100,000 scale from https://www.epa.gov/waterdata/get-nhdplus-national-hydrography-dataset-plus-data, while NHD High Resolution from https://www.usgs.gov/core-science-systems/ngp/national-hydrography is being used to create NHDPlus high resolution https://www.usgs.gov/core-science-systems/ngp/national-hydrography/nhdplus-high-resolution.

P9 L17. "Perhaps there is an opportunity to use NWM velocity forecasts ..." This seems speculative and is not followed up on. Consider either deleting it or building out the idea you had in mind.

The paragraph starting P9 L35 to the end of the paper is not conclusions drawn from results presented in the paper. As such, it should not be in the conclusions. It may be appropriate for discussion, but while certainly software considerations are important (OpenDAP, THREDDS, GUI etc), the paper has not said anything about them up to this point and tacking this discussion on at the end is a digression and distraction from the results of the study.

P2, L10-12. A citation describing how the methodology has been added to the NWC operational framework would be good.

P2, L13 (and P1, L15). Avoid claiming this paper is "the first extensive evaluation".

Some may say that there have been earlier evaluations (e.g. Zheng et al., 2018a; Zheng et al., 2018b; Godbout, 2018)

P2, L33-35. Please state how the relief between each cell and the nearest stream is calculated. If you are using the HAND layers from Liu et al. (2018) then this is computed using the TauDEM distance down function (Tesfa et al., 2011).

P3 L14-39. Section 2.2. A lot of the details here seem unnecessary. For example, the details about four forecast configurations, the products being made available to RFC's and on NOMAD do not seem relevant to the analysis reported. Further P4 L29 indicates "query the appropriate NWM output". What was the appropriate NWM output for this paper. Was it a forecast or one of the analysis and assimilation products?

P4 L4-13. Section 2.3. In contrast to section 2.2, the details about USFIMR are quite limited. It may be helpful to say a bit about the spatial resolution from the different satellite sensors and how these were rescaled for comparison with HAND inundation.

**Technical corrections**

P1 L17. I suggest delete "both" and "and". How is a quantitative comparison different from a detailed evaluation? I think this should read "These comparisons are made quantitatively through a detailed evaluation ..."

P2 L19. This sentence seems incomplete. What does  stand for?

P3 L24. In the review draft I received the dot is a light pink, not red.

P3 L30. "most valuable models for prediction". This is a subjective statement. The most valuable models depend on purpose.

P4 L10. Incorrect parentheses.

[Figure]

P4 L28. For specificity please state the URL used to download the HAND products (presumably https://web.corral.tacc.utexas.edu/nfiedata/).

P5 L16. Indicate that units on Fpred, NFpred and Fobs are area units. Just saying Fpred is predicted flood may create the impression that this is a discharge, which it is not.

P5 L34. In what sense are the HAND products "recyclable". I suggest rephrase.

P6 L7. It is conventional to introduce figures in order. Here Figure 5 is introduced before Figure 4.

P6 L33. "where" last word.

P9 L11. In areas "where" ...

**References**

Godbout, L. D., (2018), "Error assessment for Height Above the Nearest Drainage Inundation Mapping," Master of Science in Engineering Thesis, The University of Texas at Austin, https://repositories.lib.utexas.edu/handle/2152/68235.

Liu, Y. Y., D. R. Maidment, D. G. Tarboton, X. Zheng and S. Wang, (2018), "A CyberGIS Integration and Computation Framework for High-Resolution Continental-Scale Flood Inundation Mapping," JAWRA Journal of the American Water Resources Association, 54(4): 770-784, 10.1111/1752-1688.12660.

Sangwan, N. and V. Merwade, (2015), "A Faster and Economical Approach to Flood-plain Mapping Using Soil Information," JAWRA Journal of the American Water Resources Association, 51(5): 1286-1304, 10.1111/1752-1688.12306.

Tesfa, T. K., D. G. Tarboton, D. W. Watson, K. A. T. Schreuders, M. E. Baker and R. M. Wallace, (2011), "Extraction of hydrological proximity measures from DEMs using parallel processing," Environmental Modelling Software, 26(12): 1696-1709, 10.1016/j.envsoft.2011.07.018.

Zheng, X., D. R. Maidment, D. G. Tarboton, Y. Y. Liu and P. Passalacqua, (2018a), "GeoFlood: Large-Scale Flood Inundation Mapping Based on High-Resolution Terrain Analysis," Water Resources Research, 54: 10013-10033, doi:10.1029/2018WR023457.

Zheng, X., D. G. Tarboton, D. R. Maidment, Y. Y. Liu and P. Passalacqua, (2018b), "River Channel Geometry and Rating Curve Estimation Using Height above the Nearest Drainage," JAWRA Journal of the American Water Resources Association, 54(4): 785-806, doi:10.1111/1752-1688.12661.
* * *

---

## Referee Comment (RC2) · D. Blodgett (Referee) · 24 May 2019

The review comments provided by other reviewers are comprehensive and I agree on practically all points. The discussion below adds additional critique from my perspective. I suggest accepting this manuscript with minor revisions per the author's reconciliation of comments recieved.

This paper is an overview of an evaluation of the combined skill of the operational National Water Model flow estimates, synthetic rating curve transfer to stage and Height

[Figure]

Above Nearest Drainage mapping technique. It demonstrates that the NWM-HAND system of models is capable of producing semi-realistic maps of (binary – flooded or not) inundated area. Given that this is an evaluation of an entire suite of loosely coupled models, the claim that it is "comprehensive" is a stretch. To be a "comprehensive evaluation", each individual model would need to be considered in isolation and/or errors from each model / data source would need to be controlled for. A better description might be "integrated evaluation" or "combined skill evaluation".

The ancillary investigation of causes of poor evaluation results is a valuable contribution and is well thought-out and documented analysis. I would suggest being more explicit that a rough 2D analysis of peak flooded area was used to generally identify flood inundation problem areas and that the causes are investigated through identification of poor performance in specific parts of the coupled NWM-HAND system. Bring some of the limitations and realistic potential of the analysis up from the discussion into the introduction. Also bring a summary of the issues found up to the abstract and/or introductory section. Do any of the errors you found not align with what you would expect from a modeling system with the formulation of the NWM-HAND system?

The paper includes a brief introduction to the Operational National Water Model application of the WRF-Hydro-NWM software and the synthetic-rating-curve / HAND flood inundation model. The description is lacking discussion of the meteorological forcing data, soil / subsurface data, details of how NWM routing and surface storage parameters are derived, and other important aspects of the model that would affect performance and could be included in such an evaluation. The introduction is also lacking a general overview of the NWM's objectives which could/should be used to temper the expectations and focus the aims of an evaluation. e.g. The NWM is intended to predict impending high-flows for the purposes of flood warning and potential inundated area guidance.

Noting that the NWM-HAND system is not used for official forecasts and is to be considered for guidance only at this stage in its development. Given these kinds of caveats,

the evaluation presented in this manuscript is of great value as it demonstrates that the NWM-HAND system is producing flood inundation products that would be generally useful for the intended purpose. Finally, more information on the nature of the retrospective model run should be provided. Given that the NWM is only calibrated in some locations, to a small set of potential calibration targets, it should not be expected to produce realistic flow volumes. Additionally, the retrospective does not assimilate observed streamflow – which should be noted and taken into account here.

For areal comparisons, "Perennial NHD water bodies" are removed – what about "NHD Area" features that are used to denote double line streams? The water body / area features are used inconsistently in the NHD but at the end of the day, the "NHD Area" features should be considered water as they are used as a mask over top of flowlines and artificial paths.

Given that flooding is a 4D (XYZT) phenomena, using a 2D (XY) evaluation technique seems a bit limited and should be justified. Why was the evaluation not weighted using flood depth? Why was the evaluation not weighted by proximity to observed peak timing? I think there is value in the "max flood extent" evaluation, but its choice, limitations, and utility should to be discussed.

Why is the analysis binned by stream order? Given the morphology (stream density) in different parts of the country, stream order is different for streams that are equivalent on other metrics such as modeled-surface-contributing area.

It strikes me that this paper would benefit from some hypotheses based on the characteristics of the modeling approach. e.g. Given that HAND is not a physically based model in that it does not route flow over the landscape or preserve mass, we would expect small errors in stage to produce large errors in inundated areas in low-relief landscapes.

I wouldn't represent issues caused by misuse of the NHD data model in hydrologic model formulation as "errors in the NHD". While the NHD has numerous anomalous

representations of the network, it is just one source of data that must be considered when constructing a hydrologic model. For example, the decision to initiate a headwater flowline is a best guess of perennial flow and the distinction of a sink or a water body that connects to an intermittent flowline is based on the conditions considered for mapping and not extreme hydrologic scenarios encountered by the NWM. The misuse of NHD as a hard-and-fast geospatial framework for hydrologic modeling should be seen as a general flow in the NWM and not characterized as "errors" in a cartographic dataset.

---

## Author Comment (AC1) · 3 Aug 2019

Dear Editor and Reviewers,

Thank you all for the opportunity to revise this manuscript.

We want to first note that the suggestions to change error statistics and include an analysis of the bias introduced by the NWM to the NWM-HAND methods (Tarboton) were very useful but caused some changes to the format of the paper. Because of this, not all new additions / changes can be highlighted in these responses. However, we

have made every effort to specifically address each reviewers' specific concerns.

The largest structural change is that, as the methods section grew, it became more useful to move all methods into their own section and allow the results section to simply communicate what we found. A new (4.2-4.3) section describing the analysis of gaged catchments has been added and the discussion/conclusion have been updated to reflect these results.

Attached to this submission is the revised manuscript, with continuous page numbers and in-text figures.

In the remainder of this response Dr. Tarboton's requests are indicated as \*\*; Our responses are surrounded by » TEXT «; And specific sentences from the text are surrounded by parenthesis (TEXT).
* * *
Dear Dr. Tarboton,

First, thank you for the thorough and incisive review. It substantially aided the revisions of this paper and we have added you to the acknowledgements section of the paper.

\*\*Firstly, both reviewers raised concerns with the title of the paper.

»We recognize the challenge with executing a truly comprehensive evaluation as conceptualized by the reviewers. As such the title has been changed (per D. Blodgett's suggestion) to:

An Integrated Evaluation of the National Water Model (NWM) Height Above Nearest Drainage (HAND) Flood Mapping Methodology. «

\*\*Dr. Tarboton suggested changing the error statistic used to better eliminate arbitrary factors.

»Thank you for this comment and pointing out the issues with the arbitrary convex hull and how the inclusion of matching dry regions may bias our results. To address this,
we have adopted a new comparison that calculates accuracy, as well as overprediction and underprediction. These new values guide the remainder of the analysis. These are calculated by classifying the observed and simulated rasters cell-wise as WW, WD, DW, DD where W refers to wet and D refers to dry. The first character in the classification references the cell state in the observed flood map while the second refers to the state of the cell in the simulation.

Accuracy = WW / (WW + WD + DW) (fit index used Zheng 2018, and Sangwan, 2015); Over = DW / (WW + WD + DW); Under = WD / (WW + WD + DW);

These can be found in the revised manuscript as equations 3-5. This new metric did not change the overall conclusions of the tendency of NWM-HAND to under predict floodplain level inundation but did provide a more robust discussion and analysis that have improved the paper. «

**Dr. Tarboton requested that we report the matching and non-matching area between observed and modeled floods as well as total area.

»The agreement of total area (Total Simulated Wet Cells / Total Observed Wet Cells) can be seen in new figures for the flood plain analysis (Fig. 2) and for the catchment level analysis (Fig. 5). The matching and non-matching areas are represented via the Accuracy (matching), Over (non-matching) and Under (non-matching) statistics and visualized in figure 3 as a stacked bar plot and reported in table 2 and 3.

These images were added for clarity and to address this point. «

**Dr. Tarboton suggested generating flood rasters for all NHD catchments that have a USGS gage and compare them to those driven by the NWM ones to better separate out errors.

»A new section (4.2 and 4.3) was added in the revised manuscript addressing this concern for the 54 available catchments that were completely contained in a USFIMR bounding box and had a recorded NWIS and NWM-reanalysis flow values. Overall,

we found that the uncertainties in the NWM forecasts have a limited influence on the accuracy of the simulated flood extent and have documented these findings in the new sections. «

\*\*Dr. Tarboton suggested better articulating the issues with raster resolution. In addition, make figure 5A more compelling and potential problems with roughness (Manning's n), slope, and the synthetic rating curves as sources of error.

Thank you for this comment and pointing out where our prior analysis was unclear. While testing the sensitivity of the SRC Manning Equations to roughness and wetted perimeter, we discovered that our previous inclinations towards wetted perimeter being a driving factor were incorrect. In lines 298-314 we state:

(Keeping slope (NHD attribute) and the cross-sectional area required to generate a stage of 3.8 m constant, we independently varied the roughness coefficient (n) and the hydraulic radius (via the wetted perimeter), solving for a Q of 80 m3/s. In doing so we found that the SRC relationships are generally insensitive to changes in hydraulic radius (needed to be increased by a factor of 10), but were sensitive to changes in Manning's n.)

Similarly, we tested these relationships for all catchments where we had a know Stage (taken from a cross section of the HAND and USFIMR map) and Q (from NWIS)

In doing this, the most sensitive factor is roughness which is discussed at multiple points throughout the revised manuscript and highlighted in both the discussion and conclusion. For example, lines 479-483:

(An analysis of NWM-HANDs sensitivity to changes in Manning's n and cross-sectional geometries indicate that SRCs are insensitive to changes in hydraulic radius (ergo wetted perimeters) but are very sensitive to changes in Manning's n. As a general rule of thumb, the current SRCs underpredict n in lower order reaches and overpredict n in higher order reaches. In all cross-sectional geometries we tested, observed streamflow

(NWIS) stage (USFIMR cross section) relationships were achievable with a variable n, save those with zero relief.) «

**Dr. Tarboton requested a more thorough examination of what went wrong in figure 5B:

»To really understand what was going on in this instance we needed a gaged reach to better dissect whether the previous large stage resulted from poor NWM prediction or a poor SRC curve. As such we changed our analysis to look at gaged reach upstream of our last example. This new reach can be seen in Figure 7A and is discussed in lines 315-322. «

**Dr. Tarboton asked us to explicitly state which NHD versions are used:

»Thank you for this comment. The NHD version used is the medium resolution. This is now stated in line 63-65.

(For the context of this study, all references to the NHD refer to the medium resolution dataset unless otherwise stated.) «

**Dr. Tarboton asked us to remove comment on velocity or expand on its meaning:

»Thank you for identifying the isolated nature of this comment. The idea of integrating the NWM velocity has been expanded on in lines 337-348. In text:

(A second possible alternative to refactoring is to make use of the NWM velocity and flow estimates to define cross sectional areas from the NWM forecast (equation 9). The intention would be to allow the physical model (NWM) and routing-routines (WRF-Hydro) to deal with issues of volume preservation. The resulting cross-sectional areas could be used as an Area-Stage rather than Q-Stage look up within the existing SRCs. This would work around some of the issues with roughness (outsourcing to the NWM) while capitalizing on the observed accuracies in the floodplain cross sections. Moreover, by controlling for the volume of water in the channel instead of the height, low lying areas will be less prone to exaggeration. Such a change would require (A) an

understanding of how the NWM is handling hydraulics and thus velocity and (B) a test of how variations in velocity impact volume estimation. Both are interesting pursuits in their own right but out of scope for this paper.) «

**Dr Tarboton suggested moving the discussion of software from the collusion to the discussion

»Thank you for this suggestion. We have moved this section to the discussion and drastically reduced the detail. Please see lines 462-468. «

**Dr. Tarboton requested a citation of how the methodology has been added to the NWC operational framework:

»Unfortunately, we are unaware of any official citation for this. Instead we have cited the HydroShare resource for Hurricane Harvey (line 45-46). «

(NOAA National Water Center, E. Boghici, D. Arctur (2018). NOAA NWC - Harvey NWM-HAND Flood Extents, HydroShare, https://doi.org/10.4211/hs.fe85a680d0144e79b39e8c483dc1e5aa)

**Dr. Tarboton suggested removing comments of 'first extensive evaluation' comparison

»Thank you for the comment. We have noted the comment and removed all references to first extensive evaluation. Nevertheless, our analysis is novel in that it looks solely at the performance of the integrated NWM-HAND approach for a large sample of locations. «

**Dr. Tarboton requested we state how relief between cells is calculated:

»We made use of the precomputed HAND rasters and have included the TauDEM distance down function reference you provided. This is now explained in line 82-83.

(In the pre-computed HAND rasters, relief was calculated via the TauDEM distance down function (Tesfa et al., 2011)) «

\*\*Dr. Tarboton asked us to clarify what the "appropriate NWM output" means:

»Thank you for identifying this sloppy sentence. The product used was the NWM version 1.2 reanalysis product which is now explicit stated in lines 137-138

(The timestamp of each USFIMR satellite image was used to query the needed NWM v1.2 reanalysis values and generate an inundation map.) «

\*\*Dr. Tarboton asked us to add some info on USFIMR development and how rasters are aligned.

»Thank you for the interest in the USFIMR products. We have pointed to the documentation for the shapefile development (lines 128-130)

(The USFIMR web portal provides more information on each flood, the specific sensor, as well as supplementary data including NED elevation and upstream NWIS hyperlinks (http://sdml.ua.edu/usfimr).)

and have described how rasters were created and aligned in lines 141-147.

(To facilitate comparison, the USFIMR shapefiles were projected from NAD83 / Conus Albers (CRS 5070) to a WGS84 coordinate reference system (CRS 4269). For each shapefile, a clipping extent, derived as a concave hull was created to ensure that all pixels being evaluated were within the USFIMR classification bounds. A waterbody mask was created by combining the perennial NHD water bodies (NHD Fcode 39004, 39009) and NHDAreas (NHD FCode 40300, 40307, 40308, 40309) in each extent. The USFIMR flood, extent, and waterbody mask, were all rasterized to the 10m HAND grid using the fasterize R package (Ross, 2018). All cells that were not within the concave hull or covered by a waterbody mask, were set to NA prior to comparison.) «

\*\* Dr. Tarboton pointed out some technical corrections:

»Thank you for your detailed look at our paper, all suggested technical corrections have been accepted and incorporated in the revised manuscript including grammatical

correction, subjective statements, the description of red/pink. «

Again, thank you for helping make this paper substantially better than its original submission,

Sincerely,

Mike Johnson, Dinuke Munasinghe, Dami Eyelade, Sagy Cohen

Please also note the supplement to this comment:
https://www.nat-hazards-earth-syst-sci-discuss.net/nhess-2019-82/nhess-2019-82-AC1-supplement.pdf

―――――――――――――――――――――

**Supplement:**

[revised manuscript text omitted]

**2. Background**

To assess the accuracy of the NWM-HAND methodology, inundation maps were created for 28 floodplains across CONUS and compared to an observational flood map repository. Each product used in this analysis is described below.

**2.1 Height-Above-Nearest-Drainage (HAND) Method**

60 As the name implies, a Height Above Nearest Drainage (HAND) map contains the vertical distance between a location (gridcell) and its nearest stream reach. Producing a HAND map requires a terrain dataset and a spatial representation of a region's river network to define local drainages. In the USA these primarily come from the USGS National Elevation and Hydrography Datasets respectively (NED and NHD). For the context of this study, all references to the NHD refer to the medium resolution dataset unless otherwise stated. Each polyline in the NHD is described as a 'reach' and assigned a unique common identifier

65 (COMID). To calculate the 'nearest drainage', the contributing catchment for each reach is rasterized to the spatial resolution of the DEM. To illustrate this, a generic catchment (reach 101) is shown in Figure 1A. In this example, all cells in the catchment inherit the COMID (101) associated with the draining reach (Figure 1B).

Figure 1: The HAND methodological workflow: (A) The contributing catchment to a defined outlet is rasterized to the resolution of the supporting DEM (B) This raster is reclassified to match the identifying code of the outlet. (C) In and out-of-stream cells are identified using the flowline vector. The relief between all out-of-stream cells from
75 their nearest in-stream-cell is calculated to define a HAND raster. (D) From a given flow rate, a rating curve can be used to convert flow to stage. (E) A reclassification table can be built relating the reach COMID to the current

stage. (F) The table can then be used to reclassify the catchmask, into a water-leve rasterl. (G) Subtracting C from F yields a water-level above surface raster. All values less than 0 can be set to 0 and the remaining show the estimated flood height at each cell (H).

80

To generate a HAND raster, a river network is used to determine in-stream and out-of-stream cells. For all out-of-stream cells, the relief between that cell and the nearest in-stream cell is calculated (Figure 1C). In the pre-computed HAND rasters, relief was calculated via the TauDEM distance down function (Tesfa et al., 2011). Once computed, HAND raster's can be used to define hydraulic flood plain cross-sections for all reaches. The derived geometries provide cross sectional areas (A) and hydraulic radius (R) inputs for the Manning's equation (Zheng et al., 2017):

85

$$Q(y) = \frac{1}{n} \times A \times R^{\frac{2}{3}} \times S^{\frac{1}{2}}$$
(1)

where Q(y) is the discharge for a given stage, n is the Manning's roughness coefficient, and S is the reach slope. Reach slope 90 is an attribute of the NHD dataset, while a constant channel roughness (n) of 0.05 has been adopted for synthetic rating curve (SRC) generation (Zheng et al., 2018, Johnson and Coll, 2017). From this equation, streamflow requirements for a set number of stage values can be generated (Zheng et al., 2017). Finally, subtracting a HAND raster from a water-level raster yields a water height raster (Figure 1G), where any value greater than zero is classified as flood and values less than zero indicate dry cells (Figure 1H).

95

In 2017, HAND raster's and SRCs were generated for CONUS using the 10-meter NED and NHD datasets on the ROGER supercomputing system at the University of Illinois Urbana Champaign (Liu et al., 2018; Zheng et al., 2017). All products are catalogued by HUC6 (the basin level units in the WBD) on the UT Corral Server. (https://web.corral.tacc.utexas.edu/nfiedata/).

**2.2 National Water Model (NWM)**

100 The NWM serves as a cornerstone of the new NOAA Water Initiative to provide integrated predictive capabilities that promote resilience and mitigation of water risks (Cline and Maidment, n.d.). In August 2016, version 1.0 was made operational, expanding the Nation's forecasting domain from approximately 9,000 forecasting locations to 2.7 million reach outlets along the NHD. At each of these points the NWM generates hourly streamflow forecasts and 1-km forecasts of soil moisture, runoff, snow water equivalent, and other water balance states.

105

The core of the NWM is the WRF-Hydro modelling architecture supported by the National Center for Atmospheric Research (NCAR) (Gochis et al., 2018). The NWM routes water over the NHD, producing streamflow values at the end point of each reach (indexed by COMID (Figure1A – pink dot)). The NWM runs in four configurations: analysis and assimilation, short range, medium range, and long range (NOAA, 2016a; 2016b; Salas et al., 2017). The analysis and assimilation configuration

- 110 assimilates observed streamflow data from the USGS NWIS network and provides an hourly snapshots of current hydrologic conditions out to three hours. The short-range configuration produces hourly deterministic streamflow and hydrologic state forecasts 18 hours out; the medium range configuration produces 3-hourly forecasts out to 10 days; and the long-range configuration generates a 30-day, four-member forecast ensemble (NOAA, 2016a).
- 115 Operational data is publicly available for a 48-hour rolling window on the NOAA NOMADs server and can be accessed directly or through a number of user-contributed packages and end-points (Johnson, 2018a). In addition to the operational products, 23-year reanalysis studies have been run for NWM versions 1.0, 1.2, and now 2.0. These products use downscaled NLDAS-2 climate forcing's with Noah-MP, a groundwater bucket model, overland and subsurface routing, and NHDplus channel routing. Unlike the operational Analysis and Assimilation (A&A) products, the reanalysis simulations do not assimilate USGS streamflow data and have been calibrated in limited number of basins (Gochis et al., 2016). In this study, the
- v1.2 reanalysis product was accessed through Amazon Web Services (https://registry.opendata.aws/nwm-archive/).

**2.3 U.S. Flood Inundation Map Repository (USFIMR)**

[revised manuscript text omitted]

- 175 In this paper we are also interested in isolating the biases stemming from NWM streamflow forecasts and from issues with HAND/SRC layers. To do this, we collected all USGS National Water Information System (NWIS) gages within the minimum bounding box of each USFIMR case study. For each NWIS gage, the collocated COMID was identified using the Network Linked Data Index (HydroData binding). To be kept in our analysis, each catchment had to have an NWIS and NWM record for the date/hour of interest; (2) the complete catchment had to be contained within the USFIMR concave hull and; (3) a flood
- 180 needed observed in that catchment within the USFIMR map. Streamflow values from the NWIS and NWM were compared using a normalized mean error (nME) and normalized mean absolute error statistic (nMAE):

$$nME = \frac{Q_{NWIS} - Q_{NWM}}{Q_{NWIS}} \tag{6}$$

$$nMAE = \frac{ABS(Q_{NWIS} - Q_{NWM})}{Q_{NWIS}}$$
(7)

In total, these filters yielded 54 unique catchments within the 28 USFIMR floodplains for which NWM-HAND and NWIS-HAND inundation maps and AOU metrics were generated.

The changes between the NWIS and NWM-driven flood maps were calculated as the standardized difference in the accuracy (A) metric of each (equation 8). Equation 8 utilizes the percentage-based properties of the accuracy metric to describe the normalized level of change experienced by using known streamflow values. Positive values indicate the use of NWIS data increased the accuracy of the produced flood map while negative values indicate the use of NWIS decreased overall accuracy.

$$\Delta A = \frac{A_{NWIS} - A_{NWM}}{A_{NWIS}} \tag{8}$$

**4. Results**

**4.1 Area Comparison**

First, we evaluate the ability of the NWM-HAND method to produce the correct amount of inundated area. Area ratios (Eq. 195 2) are shown in Figure 2.

Figure 2: A histogram of area-ratios for our 28 floodplains show a tendency for the NWM to underpredict flood extents.

- 200 Figure 2 shows that 82% of the case studies had an area ratio less than one. Of the under predicting cases, the average arearatio was 60% (40% underprediction in area). The overall median of the sample was 73%, with an Inter Quartile Range (IQR) of 47-96%. While there are fewer cases of total-area over prediction (5), the instances tend to be large with a mean of 230.4%. This suggests that in all but the extreme cases, inundated areas are underrepresented.
- 205 From a spatially explicit perspective we evaluated how the NWM-HAND approach captures cell-level inundation. Figure 3 shows the AOU metrics for each flood as a stacked bar-plot.

---

## Author Comment (AC2) · 3 Aug 2019

Dear Editor and Reviewers,

Thank you all for the opportunity to revise this manuscript.

We want to first note that the suggestions to change error statistics and include an analysis of the bias introduced by the NWM to the NWM-HAND methods (Tarboton) were very useful but caused some changes to the format of the paper. Because of this, not all new additions / changes can be highlighted in these responses. However, we

have made every effort to specifically address each reviewers' specific concerns.

The largest structural change is that, as the methods section grew, it became more useful to move all methods into their own section and allow the results section to simply communicate what we found. A new (4.2-4.3) section describing the analysis of gaged catchments has been added and the discussion/conclusion have been updated to reflect these results.

Attached to this submission is the revised manuscript, with continuous page numbers and in-text figures.

In the remainder of this response Dr. Blodgett's requests are indicated as **; Our responses are surrounded by » TEXT «; And specific sentences from the text are surrounded by parenthesis (TEXT).
* * *
Dear Dr. Blodgett,

First, thank you for the clear and thought-provoking review. It substantially aided the revisions of this paper and we have added you to the acknowledgements section.

**Firstly, both reviewers raised concerns with the title of the paper.

»We recognize the challenge with executing a truly comprehensive evaluation as conceptualized by the reviewers. As such the title has been changed (per D. Blodgett's suggestion) to:

An Integrated Evaluation of the National Water Model (NWM) Height Above Nearest Drainage (HAND) Flood Mapping Methodology. «

**Dr. Blodgett asked us to be more upfront with our choice of using a 2D fit statistic and the implications of not treating floods as a 4D event.

»Thank you for this comment, it prompted some thoughts about what our analysis truly

entails, the choice of methods, and why it was important.

The choice for a 2D statistic (XY) is driven by the limitations of remote sensing imagery that only offers a snapshot at a single time point (T). Analysis of time-space outside of this snapshot is doable for streamflow and simulated events but not for our observed 'truth' reference dataset. As for the depth dimension, while there are some new methods for looking at flood water depths from RS imagery (see Cohen et al, 2018), doing so would have added a new source of uncertainty into an analysis where we were already trying to isolate and attribute errors from multiple sources.

In the revised manuscript we now explicitly state that we are implementing a 2D analysis of the flooded area coinciding with the timing of aerial imagery and that it should not be read that we are analyzing peak flooded areas (lines 170-174).

(The choice of a 2D fitness statistic (examining only the extent of flood, as opposed to depth and timing of flood propagation) is governed by the aerial imagery products available (which only captures the extent of the flood, at a singular point in time). By electing this form of evaluation, we only analyze the strengths of NWM-HAND simulations at the given time-step coinciding with the time of image capture (not necessarily peak flooding).) «

**Dr. Blodgett suggested bringing our discussion of limitations and realistic potential up from the discussion to the introduction/abstract.

»Thank you for this important suggestion. We have moved the ideas as suggested. Please see lines 44-49.

(The current objective of the NWM-HAND approach is high-flow prediction for the purposes of flood warning and guidance. Model accuracy should therefore be viewed in this context and expectations should be tempered while recognizing the importance of having an operational, continental scale flood forecasting system.) «

**Dr. Blodgett asked us to include more information regarding the NWM forcing data,

parameterization, and routing.

» Thank you for this suggestion. To be upfront, deciding on the level of detail to include with respect to NWM, HAND, SRC, and USFIMR background has proven to be a difficult task to making this paper both complete and concise. As such we have listed the attributes of the model you suggest but have avoided discussing any implications. Further, we point readers to a presentation talking about the model in detail (please see lines 106-113).

(The core of the NWM is the WRF-Hydro modelling architecture supported by the National Center for Atmospheric Research (NCAR) (Gochis et al., 2018). The NWM routes water over the NHD, producing streamflow values at the end point of each reach (indexed by COMID (Figure1A – pink dot)). The NWM runs in four configurations: analysis and assimilation, short range, medium range, and long range (NOAA, 2016a; 2016b; Salas et al., 2017). The analysis and assimilation configuration assimilates observed streamflow data from the USGS NWIS network and provides an hourly snapshots of current hydrologic conditions out to three hours. The short-range configuration produces hourly deterministic streamflow and hydrologic state forecasts 18 hours out; the medium range configuration produces 3-hourly forecasts out to 10 days; and the long-range configuration generates a 30-day, four-member forecast ensemble (NOAA, 2016a).) «

**Dr. Blodgett noted that the introduction is lacking a general overview of the NWM's objectives which could/should be used to temper the expectations and focus the aims of an evaluation.

» Thank you for pointing this out. We have added a caution of sorts to the introduction as well as a statement reflecting the current goals and objectives of the model (please see lines 44-49; see above).

(The current objective of the NWM-HAND approach is rapid flood inundation prediction for the purposes of disaster warning and guidance. Model accuracy should therefore

be viewed in this context and expectations should be tempered while recognizing the importance of having an operational, continental scale flood forecasting system.) «

\*\*Dr. Blodgett requested more information on the nature of the retrospective model run noting that it is only calibrated in some locations and should not be expected to produce realistic flow volumes. Additionally, he observed that the retrospective does not assimilate observed streamflow and suggested including such a remark.

»Thank you for the comment, both of these have been noted (please see lines 115-121).

(In addition to the operational products, 23-year reanalysis studies have been run for NWM versions 1.0, 1.2, and now 2.0. These products use downscaled NLDAS-2 climate forcing's with Noah-MP, a groundwater bucket model, overland and subsurface routing, and NHDplus channel routing. Unlike the operational Analysis and Assimilation (A&A) products, the reanalysis simulations do not assimilate USGS streamflow data and have been calibrated in limited number of basins (Gochis et al., 2016). In this study, the v1.2 reanalysis product was accessed through Amazon Web Services (https://registry.opendata.aws/nwm-archive/).)

\*\*Dr. Blodgett asked why we had not included NHD Areas in our masked-out regions.

»Thank you for this suggestion. We were unaware of the NHD area product and have now included it in our mask. Moreover, the NHD Fcodes (for both water bodies and areas) have been listed in text to increase transparency (see lines 144-145).

(A waterbody mask was created by combining the perennial NHD water bodies (NHD Fcode 39004, 39009) and NHDAreas (NHD FCode 40300, 40307, 40308, 40309) in each extent.) «

\*\*Dr. Blodgett aske to re consider our binning by stream order?

»Despite variable density of streams across the country there was clear evidence in our evaluations that lower order reaches underpredict flood extents while higher order

(>4) reaches performed better. We attribute this to the use of a single default Manning n coefficient in the SRC generation and discuss the implications of this in a few spots throughout the manuscript. Most relevantly, we show how stream order is a driving factor resulting in disparate tendencies at the floodplain and catchment levels. «

**Dr. Blodgett suggested generating a driving hypothesis and provided the example that "Given that HAND is not a physically based model in that it does not route flow over the landscape or preserve mass, we would expect small errors in stage to produce large errors in inundated areas in low-relief landscapes."

»Thank you for this succinct explanation of the phenomenon we were trying to describe as "volume control" in regions of low-relief. This wording has been added to lines 310-312 and help clarify our point.

(Since HAND is not a physical model, it is unable to conserve volume through space or time. In areas of low relief, where many cells have similar if not equal HAND values, small errors in stage can have disproportionate errors in inundation extent at the 10m grid cell resolution.)

**Dr. Blodgett asked us to re think the distinction of 'Errors in the NHD' as a section heading

»This point is greatly appreciated. Paragraph 2 in section 4.6 now starts:

(With respect to the streamlines it is important to recognize that the NHD was developed as a cartographic representation of the nation's waterways and using a cartographic toolset for hydrologic modelling and routing applications has inherent limitations.)

This section discusses the previously listed issues in this context. A new paragraph about the challenges with using a cartographic data as a modelling geofabric has been made in lines 364-377 with specific references to issues of refactoring catchment delineations to more compact and consistent modelling units. These have been highlighted

in modifications to figure 8.

Lastly, the section heading has been changed to better represent this and add a brief discussion of DEM resolution.

(Data Models: Use, Limitations, and Adaptions) «

Again, thank you for helping make this paper substantially better than its original submission,

Sincerely,

Mike Johnson, Dinuke Munasinghe, Dami Eyelade, Sagy Cohen

Please also note the supplement to this comment:
https://www.nat-hazards-earth-syst-sci-discuss.net/nhess-2019-82/nhess-2019-82-AC2-supplement.pdf

**Supplement:**

[revised manuscript text omitted]

55 comparing simulated events against an observational dataset.

**2. Background**

To assess the accuracy of the NWM-HAND methodology, inundation maps were created for 28 floodplains across CONUS and compared to an observational flood map repository. Each product used in this analysis is described below.

**2.1 Height-Above-Nearest-Drainage (HAND) Method**

60 As the name implies, a Height Above Nearest Drainage (HAND) map contains the vertical distance between a location (grid-cell) and its nearest stream reach. Producing a HAND map requires a terrain dataset and a spatial representation of a region's river network to define local drainages. In the USA these primarily come from the USGS National Elevation and Hydrography

Datasets respectively (NED and NHD). For the context of this study, all references to the NHD refer to the medium resolution dataset unless otherwise stated. Each polyline in the NHD is described as a 'reach' and assigned a unique common identifier

65 (COMID). To calculate the 'nearest drainage', the contributing catchment for each reach is rasterized to the spatial resolution of the DEM. To illustrate this, a generic catchment (reach 101) is shown in Figure 1A. In this example, all cells in the catchment inherit the COMID (101) associated with the draining reach (Figure 1B).

[Figure]

70

*Figure 1: The HAND methodological workflow: (A) The contributing catchment to a defined outlet is rasterized to the resolution of the supporting DEM (B) This raster is reclassified to match the identifying code of the outlet. (C) In and out-of-stream cells are identified using the flowline vector. The relief between all out-of-stream cells from*

75 *their nearest in-stream-cell is calculated to define a HAND raster. (D) From a given flow rate, a rating curve can be used to convert flow to stage. (E) A reclassification table can be built relating the reach COMID to the current*

*stage. (F) The table can then be used to reclassify the catchmask, into a water-leve rasterl. (G) Subtracting C from F yields a water-level above surface raster. All values less than 0 can be set to 0 and the remaining show the estimated flood height at each cell (H).*

80

To generate a HAND raster, a river network is used to determine in-stream and out-of-stream cells. For all out-of-stream cells, the relief between that cell and the nearest in-stream cell is calculated (Figure 1C). In the pre-computed HAND rasters, relief was calculated via the TauDEM distance down function (Tesfa et al., 2011). Once computed, HAND raster's can be used to define hydraulic flood plain cross-sections for all reaches. The derived geometries provide cross sectional areas (A) and

85 hydraulic radius (R) inputs for the Manning's equation (Zheng et al., 2017):

$$Q(y) = \frac{1}{n} \times A \times R^{\frac{2}{3}} \times S^{\frac{1}{2}} \qquad (1)$$

where Q(y) is the discharge for a given stage, n is the Manning's roughness coefficient, and S is the reach slope. Reach slope

90 is an attribute of the NHD dataset, while a constant channel roughness (n) of 0.05 has been adopted for synthetic rating curve (SRC) generation (Zheng et al., 2018, Johnson and Coll, 2017). From this equation, streamflow requirements for a set number of stage values can be generated (Zheng et al., 2017). Finally, subtracting a HAND raster from a water-level raster yields a water height raster (Figure 1G). where any value greater than zero is classified as flood and values less than zero indicate dry cells (Figure 1H).

95

In 2017, HAND raster's and SRCs were generated for CONUS using the 10-meter NED and NHD datasets on the ROGER supercomputing system at the University of Illinois Urbana Champaign (Liu et al., 2018; Zheng et al., 2017). All products are catalogued by HUC6 (the basin level units in the WBD) on the UT Corral Server. (https://web.corral.tacc.utexas.edu/nfiedata/).

**2.2 National Water Model (NWM)**

100 The NWM serves as a cornerstone of the new NOAA Water Initiative to provide integrated predictive capabilities that promote resilience and mitigation of water risks (Cline and Maidment, n.d.). In August 2016, version 1.0 was made operational, expanding the Nation's forecasting domain from approximately 9,000 forecasting locations to 2.7 million reach outlets along the NHD. At each of these points the NWM generates hourly streamflow forecasts and 1-km forecasts of soil moisture, runoff, snow water equivalent, and other water balance states.

105

The core of the NWM is the WRF-Hydro modelling architecture supported by the National Center for Atmospheric Research (NCAR) (Gochis et al., 2018). The NWM routes water over the NHD, producing streamflow values at the end point of each reach (indexed by COMID (Figure1A – pink dot)). The NWM runs in four configurations: analysis and assimilation, short

range, medium range, and long range (NOAA, 2016a; 2016b; Salas et al., 2017). The analysis and assimilation configuration
assimilates observed streamflow data from the USGS NWIS network and provides an hourly snapshots of current hydrologic
conditions out to three hours. The short-range configuration produces hourly deterministic streamflow and hydrologic state
forecasts 18 hours out; the medium range configuration produces 3-hourly forecasts out to 10 days; and the long-range
configuration generates a 30-day, four-member forecast ensemble (NOAA, 2016a).

Operational data is publicly available for a 48-hour rolling window on the NOAA NOMADs server and can be accessed
directly or through a number of user-contributed packages and end-points (Johnson, 2018a). In addition to the operational
products, 23-year reanalysis studies have been run for NWM versions 1.0, 1.2, and now 2.0. These products use downscaled
NLDAS-2 climate forcing's with Noah-MP, a groundwater bucket model, overland and subsurface routing, and NHDplus
channel routing. Unlike the operational Analysis and Assimilation (A&A) products, the reanalysis simulations do not
assimilate USGS streamflow data and have been calibrated in limited number of basins (Gochis et al., 2016). In this study, the
v1.2 reanalysis product was accessed through Amazon Web Services (https://registry.opendata.aws/nwm-archive/).

**2.3 U.S. Flood Inundation Map Repository (USFIMR)**

Following the release of the NWM, academic partners at the University of Alabama developed the US Flood Inundation Map
Repository (USFIMR) to provide inundation maps for past U.S. flood events. These maps were derived by image classification
techniques from a number of satellite sensors (e.g. Landsat, Sentinel-1, 2) with some ground truthing based on secondary
sources (e.g. news reports, social media). Such maps are useful for model calibration, validation, and flood susceptibility
assessment (Cohen et al., 2018; Munasinghe et al., 2018). The USFIMR web portal provides more information on each flood,
the specific sensor, as well as supplementary data including NED elevation and upstream NWIS hyperlinks
(http://sdml.ua.edu/usfimr). A catalogue of the USFIMR maps used in this study can be found in the appendix table 1. Here it
is noted that the FloodID assigned to each flood in this analysis is consistent with those used in the USFIMR and not fully
sequential.

**3. Methods**

For each USFIMR map, the AOI and HydroData R packages were used to determine the minimum bounding box of each
floodplain and subset the NHD (Johnson, 2018a, Johnson, 2018b). A list of COMIDs and HUC6 identifiers were extracted
from the NHD attributes. For each HUC6, the HAND, catchment and rating curve products were downloaded cropped and,
when necessary, mosaicked. The timestamp of each USFIMR satellite image was used to query the needed NWM v1.2
reanalysis values and generate an inundation map. This process was repeated for each USFIMR flood map and the process is
formalized in the Flood Mapping R package (Johnson, 2019).

To facilitate comparison, the USFIMR shapefiles were projected from NAD83 / Conus Albers (CRS 5070) to a WGS84 coordinate reference system (CRS 4269). For each shapefile, a clipping extent, derived as a concave hull was created to ensure that all pixels being evaluated were within the USFIMR classification bounds. A waterbody mask was created by combining the perennial NHD water bodies (NHD Fcode 39004, 39009) and NHDAreas (NHD FCode 40300, 40307, 40308, 40309) in each extent. The USFIMR flood, extent, and waterbody mask, were all rasterized to the 10m HAND grid using the fasterize R package (Ross, 2018). All cells that were not within the concave hull or covered by a waterbody mask, were set to NA prior to comparison.

After processing, grid-cells in the simulated and observed maps for each case study were intersected and classified into four categories seen in Table 1. In these categories, 'W' refers to a wet cell and 'D' refers to a dry cell. The first letter in each category refers to the state of the cell in the USFIMR map while the second refers to the state of the cell in the NWM-HAND simulation. With this classification, four metrics were established to describe the area ratio, and rate of accuracy, overprediction and underprediction. The last three of these are derivatives of the fit statistic used by Sangwan and Merwade (2015).

*Table 1: Flood Comparison Confusion Matrix.*

|  | **Modeled Wet** | **Modeled Dry** |
|---|---|---|
| **Observed Wet** | *WW (true wet)* | *WD (false dry)* |
| **Observed Dry** | *DW (false wet)* | *DD (true dry)* |

The spatially-agnostic Area-Ratio helps establish if the NWM-HAND method is inundating the correct number of cells regardless of spatial accuracy. Any value less than one indicates the NWM-HAND method is not inundating enough cells while a value greater than 1 indicates too many cells were inundated.

$$AreaRatio = \frac{Total\ Wet\ Simulation\ Cells}{Total\ Wet\ Observed\ Cells} \tag{2}$$

Overall Accuracy is quantified as the number of cells that were correctly identified as wet (WW) divided the number of all wet cells in the simulated and observed rasters:

$$Accurate\ (A) = \frac{WW}{WW+WD+DW} \tag{3}$$

Similarly, areas of Underprediction are quantified as:

$$Under\ (U) = \frac{WD}{WW+WD+DW} \tag{4}$$

And areas of Overprediction as:

$$Over\ (O) = \frac{DW}{WW+WD+DW} \tag{5}$$

The use of a common denominator assures the summation of Over, Under and Accurate equals 1, and collectively these values form the Accurate, Over and Underpredict (AOU) metrics. The choice of a 2D fitness statistic (examining only the extent of flood, as opposed to depth and timing of flood propagation) is governed by the aerial imagery products available (which only captures the extent of the flood, at a singular point in time). By electing this form of evaluation, we only analyze the strengths of NWM-HAND simulations at the given time-step coinciding with the time of image capture (not necessarily peak flooding).

In this paper we are also interested in isolating the biases stemming from NWM streamflow forecasts and from issues with HAND/SRC layers. To do this, we collected all USGS National Water Information System (NWIS) gages within the minimum bounding box of each USFIMR case study. For each NWIS gage, the collocated COMID was identified using the Network Linked Data Index (HydroData binding). To be kept in our analysis, each catchment had to have an NWIS and NWM record for the date/hour of interest; (2) the complete catchment had to be contained within the USFIMR concave hull and; (3) a flood needed observed in that catchment within the USFIMR map. Streamflow values from the NWIS and NWM were compared using a normalized mean error (nME) and normalized mean absolute error statistic (nMAE):

$$nME = \frac{Q_{NWIS} - Q_{NWM}}{Q_{NWIS}} \tag{6}$$

$$nMAE = \frac{ABS(Q_{NWIS} - Q_{NWM})}{Q_{NWIS}} \tag{7}$$

In total, these filters yielded 54 unique catchments within the 28 USFIMR floodplains for which NWM-HAND and NWIS-HAND inundation maps and AOU metrics were generated.

The changes between the NWIS and NWM-driven flood maps were calculated as the standardized difference in the accuracy (A) metric of each (equation 8). Equation 8 utilizes the percentage-based properties of the accuracy metric to describe the normalized level of change experienced by using known streamflow values. Positive values indicate the use of NWIS data increased the accuracy of the produced flood map while negative values indicate the use of NWIS decreased overall accuracy.

$$\Delta A = \frac{A_{NWIS} - A_{NWM}}{A_{NWIS}} \tag{8}$$

**4. Results**

**4.1 Area Comparison**

First, we evaluate the ability of the NWM-HAND method to produce the correct amount of inundated area. Area ratios (Eq. 2) are shown in Figure 2.

[Figure]

*Figure 2: A histogram of area-ratios for our 28 floodplains show a tendency for the NWM to underpredict flood extents.*

Figure 2 shows that 82% of the case studies had an area ratio less than one. Of the under predicting cases, the average area-ratio was 60% (40% underprediction in area). The overall median of the sample was 73%, with an Inter Quartile Range (IQR) of 47-96%. While there are fewer cases of total-area over prediction (5), the instances tend to be large with a mean of 230.4%. This suggests that in all but the extreme cases, inundated areas are underrepresented.

From a spatially explicit perspective we evaluated how the NWM-HAND approach captures cell-level inundation. Figure 3 shows the AOU metrics for each flood as a stacked bar-plot.

[Figure]

*Figure 3: Stacked bar plot of the Accuracy, Over, and Under predict (AOU) metrics for each case study. Each bar represented 100% of the inundated cells in the intersected simulation and observed case study and the percentage of cells that were accurate, over and under predicated.*

Figure 3 emphasizes the tendency towards underprediction (maroon) across our case studies. Table 2 shows that, of all flooded cells (in either the simulation or observed data), 37 - 63% (IQR) fall within the under predicted category, 10 - 36% (IQR) were overpredicted and 10-27% (IQR) are correctly identified with a mean of 19%.

*Table 2: Summary statistics across the 28 case studies.*

|  | Min | 1st Quartile | Median | Mean | 3rd Quartile | Max |
|---|---|---|---|---|---|---|
| **Accurate** | 0 | 0.10 | 0.15 | 0.19 | 0.27 | 0.53 |
| **Over** | 0 | 0.10 | 0.25 | 0.27 | 0.36 | 0.76 |
| **Under** | 0.09 | 0.37 | 0.47 | 0.53 | 0.63 | 1.0 |

These statistics show a clear signal that the NWM-HAND method is limited in its ability to capture inundation and is almost twice as likely to underpredict than overpredicted the status of a missed cell. Noting the stated intention of NWM-HAND is creating an operational flood mapping framework for emergency guidance, this tendency raises concerns. These results indicate there is significant room for improvement in the NWM-HAND inputs but making these improvements requires a better understanding of the largest sources of error. In the following sections we more closely explore the 54 gaged catchments to better understand the influences driving performance.

**4.2 Catchment Area Comparison**

To evaluate the influence of NWM uncertainties on the inundation, summary statistics for the NWM and NWIS-driven AOU metrics are presented in Table 3.

*Table 3: IQR and mean values for NWM-NWIS flow comparisons and inundation metrics for 54 NHD catchments.*

| | 1st quartile $A_{NWM}$ / $A_{NWIS}$ | Median $A_{NWM}$ / $A_{NWIS}$ | Mean $A_{NWM}$ / $A_{NWIS}$ | 3rd quartile $A_{NWM}$ / $A_{NWIS}$ |
|---|---|---|---|---|
| Accurate | 9% / 9% | 16% /16% | 25% / 25% | 38% / 43% |
| Over | 17% / 23% | 45% /47% | 44% / 47% | 66% / 70% |
| Under | 4% / 3% | 26% / 21% | 31% / 28% | 53% / 46% |

Table 3 highlights that using known streamflow values generates minimal changes in average accuracy except in the upper quartiles. In a broad sense, the influence of NWM streamflow appears small (e.g. ±3%) under average conditions increasing to about ±6% on the outer quartiles. This indicates that either the NWM is preforming well at these stations or the NWM-HAND model is robust to errors in NWM input.

A linear regression between each NWM and NWIS AOU metric shows $R^2$ values of 93%, 82%, and 75% respectively. This means that approximately 7% of the variation seen in accurate prediction can be attributed to inaccuracies in the NWM forecasts, as well as 18% of the variation in over prediction and 25% of the variation in underprediction. Given the nME of the NWM forecasts throughout our catchment sample was -20% (IQR (-55)-5%), the relative order of NWM influences seem correct, while highlighting the generally small influence in overall attribution. Table 3 also indicates that, using both NWM and NWIS streamflow, the tendency at the catchment level is to over predict (NWM 17-66% IQR) rather than under predict (4-53%) inundation. This is a opposite pattern seen when looking at the entire floodplain maps.

**4.3    Influences of NWM streamflow inputs**

Of our 54 samples, the NWM produces a nMAE of 20% or less in 37% of our catchments and an nMAE of 40% or less in 50% of the catchments. Seventeen (17%) of the catchments showed a nMAE greater than 70%, and the average nME was -

20% (IQR -54 – 6%) indicating a general tendency for the NWM to underpredict high flows. To better understand the relative
bias introduced by the NWM inputs we grouped our 54 NHD catchments by their nMAE.

*Table 4: AOU metrics grouped by nMAE*

| nMAE range | Gages (cumulative %) | Mean $\Delta A$ | Absolute mean $\Delta A$ | NWIS mean A / O / U | NWM mean A / O / U |
|---|---|---|---|---|---|
| 0 - 10% | 9 (16.67%) | -0.24 | 0.68 | 25 / 47 / 29 | 25 / 47 / 29 |
| 10 - 20% | 11 (37%) | 1.64 | 1.91 | 20 / 61 / 19 | 20 / 61 / 19 |
| 20 - 30% | 4 (44%) | 3.92 | 4.18 | 21 / 36 / 43 | 21 / 32 / 47 |
| 30 - 40% | 3 (50%) | -6.36 | 6.36 | 30 / 20 / 50 | 30 / 19 / 51 |
| 40 - 50% | 3 (55.8%) | -0.3 | 6.81 | 38 / 48 / 14 | 39 / 43 / 18 |
| 50 - 60% | 6 (66.7%) | -0.55 | 8.39 | 47 / 48 / 5 | 47 / 44 / 9 |
| 60 - 70% | 8 (81.49%) | 4.90 | 31.77 | 22 / 50 / 28 | 20 / 42 / 38 |
| 70 - 80% | 1 (83.3%) | 69.52 | 69.52 | 2 / 8 / 90 | 0 / 3 / 97 |
| 80 - 90% | 3 (88.90%) | 30.15 | 30.15 | 22 / 36 / 42 | 14 / 23 / 63 |
| 90 - 100% | 2 (92.60%) | 3.54 | 84.36 | 10 / 67 / 23 | 13 / 32 / 54 |
| >100% | 4 (100%) | -360.51 | 391.42 | 12 / 46 / 41 | 12 / 74 / 14 |

For each set of basins grouped by nMAE, the number of gages, mean and absolute mean $\Delta A$ along with the average AOU
metrics for the NWM and NWIS driven maps are shown in Table 4. Looking at the mean absolute error between the NWIS
and NWM maps, there is evidence that the influence of the NWM on NWM-HAND accuracies are minimal until nMAE
exceeds 60% when there is a clear jump in the absolute mean $\Delta A$ from less than a 10% relative loss to more than 30%.

**4.3.1 NWM-HAND Errors when nMAE exceeds 60%**

When the NWM exceeds nMAE of 60%, predictable patterns of error occur. Figure 4A (Flood ID 4), shows a case where
NWM-HAND under predicts inundation due to an underpredicted NWM forecast. Figure 4B (Flood ID 9) shows a case where
an over predicted NWM forecast results in over predicted flood extents.

[Figure]

*Figure 4: Prediction errors due to NWM flow forecast: (A) NWM underprediction results in underpredicted flood extents. (B NWM overprediction results in overpredicted flood extent.*

**4.3.2 NWM-HAND Errors when nMAE is less then 60%**

Previously we noted that NWM-HAND maps showed minimal deviation from those produced with NWIS values when the NWM was within 60% of the observed streamflow. Also, we noted the catchment sample shows the opposite pattern of the floodplain analysis – errors tend towards over prediction rather than under prediction, despite the average tendencies of the NWM to under predict streamflow (nME of -20%). To visualize this, the area ratios for our 54 catchments are plotted in Figure 5.

[Figure]

*Figure 5: A histogram of area-ratios for 54 catchments show a tendency for the NWM to slightly overpredict flood extents in most cases and a long right tail of extreme over prediction.*

When compared to Figure 2 (area-ratios of floodplain case studies) we see the peak in the density curve now sits much closer to 1 with a more equal proportions on either side. This indicates that at the catchment level, NWM-HAND is better able to capture the total area of inundation. However, we see that the over predicted cases are much more extreme causing the right tail to elongate.

Of the 54 catchments, 90% were of a Strahler stream order of 4th or higher. In comparison, when all catchments in each of our USFIMR floodplains were aggregated, low order reaches made up, on average, 81% of the total stream networks. From here on out, lower order reaches refer to those with a Strahler order of 4 or less, and higher order refers to those with a Strahler order or 4 or higher. In many ways, this makes our sample catchments a biased sample favouring high order reaches. Our hypothesis here, is that the difference in patterns seen between the floodplains and catchments can be explained by the difference in the sample and population. That is low order reaches tend to be underpredicted while higher order reaches perform better with a tendency towards over prediction.

**4.4 Underprediction in lower order reaches**

A visual analysis of our floodplains also reveals a distinct pattern of under prediction in lower order reaches. We illustrate this using the only gaged 2nd order stream in our sample (Fig 6, Flood ID 8 USGS Station ID 8068325). At this reach, the NWM estimated 50 m³/s flowing through the channel producing a SRC stage of 1.82 m. The recorded NWIS flow is 80 m³/s (nMAE = 0.38) which would produce an SRC stage of 2.43 m. A cross section was made across the HAND raster for this reach and shows a stage of 3.3 is required to inundate the right bank and 3.8 m is needed to inundate the left bank to the levels seen in the USFIMR.

[Figure]

*Figure 6: Inaccurate synthetic rating curve relationship causes underprediction in a 2nd order stream*

In this case, even if the NWM was able to accurately predict streamflow, water would have remained confined to the channel. Recognizing a mismatch between the known Q (80 m³/s) and the stage observed in the USFIMR (3.8 m) we explored the assumptions driving the Manning's Equation SRC for this reach (equation 1). Keeping slope (NHD attribute) and the cross-sectional area required to generate a stage of 3.8 m constant, we independently varied the roughness coefficient (n) and the hydraulic radius (via the wetted perimeter), solving for a Q of 80 m³/s. In doing so we found that the SRC relationships are generally insensitive to changes in hydraulic radius (needed to be increased by a factor of 10), but were sensitive to changes in Manning's n. Specifically, we found that an N of 0.16 was needed to achieve a realistic streamflow-stage relationship at this reach. Based on this representative example we suggest that systematically increasing roughness in lower order reaches may help mitigate underprediction. The literature and other model structures also support such a change. For example, the NWM itself uses n values of 0.06 for first and second order reaches, and 0.055 for third and fourth order reaches. Other projects apply n values by stream order more similar to what we found in our example (n = 0.12 for second order reaches; Li, 2016). More research is needed on the relative role of roughness in the NWM-HAND method and the best way to optimize it across different stream orders and geographies.

**4.5 Overprediction in areas of low-relief**

Throughout our evaluation, over prediction in low-relief catchments became evident. Moreover, such catchments account for the majority of the long right tail seen in Figure 5. We found two type of cross-sections occurring in areas of low relief, ones that are very sensitive to errors in stage, and, ones that are fundamentally limited by the terrain representation in the HAND layer. Since HAND is not a physical model, it is unable to conserve volume through space or time. In areas of low relief, where many cells have similar if not equal HAND values, small errors in stage can have disproportionate errors in inundation extent at the 10m grid cell resolution. The Washita river at Anadarko OK case study (Figure 7A) shows a 6-order stream with low relief. The NWM estimates 473 m³/s producing and SRC stage of 3.81 m. Comparatively, the recorded flow was 495 m³/s producing an SRC stage of 3.88 m. The nMAE of 4% suggests the source of overprediction originates in the terrain layer or SRC. A cross-section of this landscape reveals a channel with relatively uniform overbank terrain. For this reach, we optimized the SRC for the cross section at an observed stage of 1.2 solving for a Q of 495 m³/s. To achieve this, n was reduced to 0.005, effectively increasing the carrying capacity of the river. This solution supports the pattern of decreasing n in higher order rivers and this emphasizes that fact that in areas with low relief, small errors in stage and or discharge can result in extreme overpredictions.

[revised manuscript text omitted]

**5. Discussion**

At the floodplain resolution, NWM-HAND is able to accurately capture 10 - 27% of inundation (IQR) with a larger tendency to under (37 - 63%) then over predict (10 - 36%). To better understand how model performance is driven by NWM, we grouped

420   a sample of 54 gaged catchments by relative NWM performance and found the NWM-HAND method is relatively insensitive to biases in NWM streamflow prediction accuracy - losing less than 10% relative accuracy. However, when the normalized mean error between observed and NWM predicted streamflow exceeds 60% the bias in inundation prediction is increased considerably. Overall, 66.7% of the catchments in our sample exhibited a NWM prediction accuracy with a nMAE less than 60%. In this paper we used a reanalysis product which was created with limited calibration and no data assimilation. When

425   the NWM-HAND is used operationally, much of the data will come from the analysis and assimilation configuration which assimilates NWIS gage data and has had a more robust calibration. In theory this should allow a greater percent of reaches to fall within the 60% nMAE threshold.

In addition to errors in streamflow magnitude, NWM timing errors can introduce issues related to flood inundation forecasting

430   and its evolution. Even though we did not explicitly study the ability of NWM-HAND to capture the temporal evolution of flood extents in this study, a few instances where the NWM improperly estimated the receding limb of the hydrograph, showed the inability of NWM-HAND to capture ponded and receding waters leading to issues of under and over prediction respectively.

435   We saw a larger tendency to overpredict (17% - 63% NWM) then underpredict (4 - 53%) inundation at the catchment level - the opposite pattern seen at the resolution of the floodplain. We argue that the tendency to underpredict area at the floodplain resolution was an aggregate result of consistently underpredicting many, small lower order catchments. From this we concluded that the base assumptions in the SRC generation was not adequately capturing lower order streams. Based on our tests, we found the SRCs are highly sensitive to the roughness coefficient used and insensitive to changes in floodplain cross-

440   section. As a general rule we believe the default Manning's coefficient used by HAND is too small in low order reaches and are arguably too high in higher order reaches. While we only tested this on the small sample of available catchments, the theory behind the Manning values, and the consistent propensity to under predict low order reaches would support such a change. Moreover, in all of our tests the existing cross-sections geometries could be altered to provide correct Q when varying

Manning's n within a range of 0.0001 and 0.2. This leads us to suggest that further work with respect to the Manning's n values used should be prioritized

Another reoccurring error in our simulations was over prediction in catchments with low relief. Such catchments allow incremental changes in stage to have large impacts on inundation extent. We suggest that a refactoring of the NHD to smaller more uniform units, and a consideration for how volume rather than stage may be used to fill catchments, may be possible solutions. Moreover, the former offers some solutions to issues arising from treating the cartographic NHD as a modelling infrastructure. That said, neither of these are trivial to accomplish or validate.

While the overall NWM-HAND methods certainly have a long way to go, we contend the results demonstrate that the NWM-HAND approach of generating flood inundation maps is useful for general guidance and risk identification but may not be suitable for pixel level analysis, resource allocation, or risk-communication.

Despite the limited accuracy found in this study, the NWM-HAND is quite an achievement that should not be discounted. The fact that an uncalibrated continental-scale model can be used to rapidly generate flood inundation maps for all case studies with a single framework (code) is of great value. More importantly, the base layers and conceptual framework underpinning the model offer the research community a resource to improve, modify, and manipulate.

Looking towards future research and development; the inevitable next version of the base layers; and operational use, issues of availability arise. Currently, accessing and combining the needed NWM-HAND products is cumbersome, and for regions straddling one or more HUC6 can be data and processing intensive. While services like the Flood Mapping R package can streamline some of the steps, disseminating data by HUC6 is not a convenient choice for users. Instead HAND products could be distributed via a web service like those used to distribute climate data and NWM gridded output or built into services like the CUAHSI sub setter (CUASHI, 2019). With workable foundations now in place, engaging with the communities that will use these products for research, map creation, and ultimately in the field seems like a next step worth considering.

**6. Conclusion**

This study offers a high-level evaluation of the confidence we can place in the operational NWM-HAND forecasts. In its current state the NWM-HAND methods have limited ability to accurately capture inundation and its skill is more constrained by the terrain and SRC inputs then NWM accuracy.

475 At floodplain level, NWM-HAND tends to underestimate flooding (82% of case studies). Of these, the mean likelihood of underpredicting a missed cell (53%) is twice that of overprediction (27%). At the catchment scale, NWM-HAND was better able to capture the total area of inundation (improvement of a mean of 6% in comparison to the floodplain level) but was more likely to overpredict (44%) than underpredict (31%) missed cells. We attribute this disparity to systemic underprediction in lower order reaches. An analysis of NWM-HANDs sensitivity to changes in Manning's n and cross-sectional geometries

480 indicate that SRCs are insensitive to changes in hydraulic radius (ergo wetted perimeters) but are very sensitive to changes in Manning's n. As a general rule of thumb, the current SRCs underpredict n in lower order reaches and overpredict n in higher order reaches. In all cross-sectional geometries we tested, observed streamflow (NWIS) stage (USFIMR cross section) relationships were achievable with a variable n, save those with zero relief.

485 We further investigated the level of bias in NWM-HAND maps coming from the NWM forecasts and found the method to be relatively insensitive until the normalized absolute error between NWM and observed flow exceeded 60%. Above this threshold, predictable patterns of large under or over prediction occur in the direction of the NWM miss.

Lastly, over prediction in large catchments with low relief is a common error. Aside from integrating higher resolution terrain

490 data, two possible solutions for remedying these systemic errors include refactoring the catchment areas used to more uniform and compacts units, and a consideration of how NWM cross-sectional area forecasts (Streamflow / Velocity) can be used to better control the spread of water in areas of low relief.

[revised manuscript text omitted]

---

## Author Response (AR1)

First, we would like to thank both reviewers for the thorough and incisive reviews, and the editor for allowing this paper to go through with minor revisions.

- Both reviewers raised **concerns with the title of the paper.** We recognize the challenge with executing a truly comprehensive evaluation as conceptualized by the reviewers. As such the title has been changed (per D. Blodgett's suggestion) to: **An Integrated Evaluation of the National Water Model (NWM) Height Above Nearest Drainage (HAND) Flood Mapping Methodology.**

- Moreover, per the suggestions of Dr. Tarboton, a more exhaustive look at HAND performance in gaged catchments has been added. In some ways this new analysis changed the way we discuss our outcomes and the format/flow of the paper. **Because of this, not all new additions / changes can be highlighted in this response. However, we have made every effort to specifically address each specific concern.**

- The largest structural change induced by reviewer comments was that all methods were moved into their own section allowing results to simply communicate what we found.

Attached to this submission please find the revised manuscript, with continuous page number and in-text figures and captions.

**David Tarboton**

**Dr. Tarboton suggested changing the error statistic used to better eliminate arbitrary factors.**

Thank you for this comment and pointing out the issues with the arbitrary convex hull and how the inclusion of matching dry regions may bias our results. To address this, we have adopted a new comparison that calculates accuracy, as well as overprediction and underprediction. These new values guide the remainder of the analysis. These are calculated by classifying the observed and simulated rasters cell-wise as WW, WD, DW, DD where W refers to wet and D refers to dry. The first character in the classification references the cell state in the observed flood map while the second refers to the state of the cell in the simulation.

> Accuracy = WW / (WW + WD + DW) (fit index used Zheng 2018, and Sangwan, 2015)
> Over = DW / (WW + WD + DW)
> Under = WD / (WW + WD + DW)

These can be found in the revised manuscript as equations 3-5. This new metric did not change the overall conclusions of the tendency of NWM-HAND to under predict floodplain level inundation but did provide a more robust discussion and analysis that have improved the paper.

**Dr. Tarboton requested that we report the matching and non-matching area between observed and modeled floods as well as total area.**

The agreement of *total area* (Total Simulated Wet Cells / Total Observed Wet Cells) can be seen in new figures for the flood plain analysis (Fig. 2) and for the catchment level analysis (Fig. 4). The *matching and non-matching areas* are represented via the Accuracy (matching), Over (non-matching) and Under

(non-matching) statistics and visualized in figure 3 as a stacked bar plot and reported in table 2. These images were added for clarity and to address this point.

**Generate flood rasters for all NHD catchments that have a USGS gage. Compare these to the NWM ones to better separate out errors.**

A new section (4.2 – 4.4) was added in the revised manuscript addressing this concern for the 54 available catchments that were completely contained in a USFIMR bounding box and had a recorded NWIS and NWM-reanalysis flow values. Overall, we found that the uncertainties in the NWM forecasts have a limited influence on the accuracy of the simulated flood extent and have documented these findings in the new sections.

**Better articulate the issues with raster resolution. Make figure 5A more compelling. Problems with roughness (Manning's n), slope, and the synthetic rating curve are all potential causes.**

Thank you for this comment and pointing out where our prior analysis was unclear. While testing the sensitivity of the SRC Manning Equations to roughness and wetted perimeter we discovered that our previous inclinations towards wetted perimeter being a driving factor were incorrect. In text (lines 376-380) we state:

>> Keeping slope (NHD attribute) and the cross-sectional area required to generate a stage of 3.8 m constant, we independently varied the roughness coefficient (N) and the hydraulic radius (via the wetted perimeter), solving for a Q of 80 $m^3$/s. In doing so we found that the SRC relationships are generally insensitive to changes in hydraulic radius (needed to be increased by a factor of 10), but were sensitive to changes in Manning's N. <<

In fact, the geometries that we tested could all generate proper discharge values when varying N between 0.001 and 0.2. Instead the most sensitive factor is that of roughness which is discussed at multiple points throughout the revised manuscript.

**The reviewer requested a more thorough examination of what went wrong in figure 5B:**

To really understand what was going on in this instance we needed a gaged reach to better dissect whether the previous large stage resulted from poor NWM prediction or a poor SRC curve. As such we changed our analysis to look at gaged reach upstream of our last example. This new reach can be seen in Figure 7A and is discussed in lines 398-404.

**Explicitly state which NHD versions are used:**

Thank you for this comment. The NHD version used is the medium resolution. This is now stated in line 112 - 114.

>>In 2017, HAND raster's and SRCs were generated for CONUS using the 10-meter NED and medium resolution NHD datasets on the ROGER supercomputing system at the University of Illinois Urbana Champaign (Liu et al., 2018; Zheng et al., 2017). <<

**Remove comment on velocity or expand**

Thank you for identifying the isolated nature of this comment. The idea of integrating the NWM velocity has been expanded on in lines 427-440. In text:

*"A second possible alternative to refactoring is to make use of the NWM velocity and flow estimates to define cross sectional areas from the NWM forecast (equation 9). The intention would be to allow the physical model (NWM) and routing-routines (WRF-Hydro) to deal with issues of volume preservation. The resulting cross-sectional areas could be used as an Area-Stage rather than Q-Stage look up within the existing SRCs. This would work around some of the issues with roughness (outsourcing to the NWM) while capitalizing on the observed accuracies in the floodplain cross sections. Moreover, by controlling for the volume of water in the channel instead of the height, low lying areas will be less prone to exaggeration. Such a change would require (A) an understanding of how the NWM is handling hydraulics and thus velocity and (B) a test of how variations in velocity impact volume estimation. Both are interesting pursuits in their own right but out of scope for this paper."*

**Move discussion of software ect from collusion to discussion**

Thank you for this suggestion. We have moved this section to the discussion and drastically reduced the detail. Please see lines 539-547.

**Add a citation of how the methodology has been added to the NWC operational framework:**

Unfortunately, we are unaware of any official citation for this. Instead we have cited the HydroShare resource for Hurricane Harvey (line 47-48).

*NOAA National Water Center, E. Boghici, D. Arctur (2018). NOAA NWC - Harvey NWM-HAND Flood Extents, HydroShare, https://doi.org/10.4211/hs.fe85a680d0144e79b39e8c483dc1e5aa*

**Remove comments of 'first extensive evaluation' comparison**

Thank you for the comment. We have noted the comment and removed all references to first extensive evaluation. Nevertheless, our analysis is novel in that it looks solely at the performance of the integrated NWM-HAND approach for a large sample of locations.

**State how relief between cells is calculated:**

We made use of the precomputed HAND rasters and have included the TauDEM distance down function reference you provided. This is now explained in line 113-114.

*In the pre-computed HAND rasters, relief was calculated via the TauDEM distance down function (Tesfa et al., 2011).*

**Identify the "appropriate NWM output"**

Thank you for identifying this sloppy sentence. The product used was the NWM version 1.2 reanalysis product which is now explicit stated in lines 164-166

*The timestamp of each USFIMR satellite image was used to query the needed NWM v1.2 reanalysis values by COMID and generate an inundation map using the HAND methodology (section 2.1).*

**Add some info on USFIMR development and how rasters are aligned.**

Thank you for the interest in the USFIMR products. We have pointed to the documentation for the shapefile development (lines 143-145)

*"The USFIMR web portal provides more information on each flood, the specific sensor, as well as supplementary data including NED elevation and upstream NWIS hyperlinks (http://sdml.ua.edu/usfimr)."*

and have described how rasters were created and aligned section 3.1.

**Technical corrections:**

Thank you for your detailed look at our paper, all suggested technical corrections have been accepted and incorporated in the revised manuscript including grammatical correction, subjective statements, the description of red/pink.

**David Blodgett**

**Dr. Blodgett asked us to be more upfront with our choice of using a 2D fit statistic and the implications of not treating floods as a 4D event.**

Thank you for this comment, it prompted some thoughts about what our analysis truly entails, the choice of methods, and why it was important.

*The choice for a 2D statistic (XY) is driven by the limitations of remote sensing imagery that only offers a snapshot at a single time point (T). Analysis of time-space outside of this snapshot is doable for streamflow and simulated events but not for our observed 'truth' reference dataset. As for the Depth dimension, while there are some new methods for looking at flood water depths from RS imagery (see Cohen et al, 2018), doing so would have added a new source of uncertainty into an analysis where we were already tried to isolate and attribute errors from multiple sources.*

In the revised manuscript we now explicitly state that we are implementing a 2D analysis of the flooded area coinciding with the timing of Aerial imagery and that it should not be read that we are analyzing peak flooded areas (lines 218-223).

*The choice of a 2D fitness statistic (examining only the extent of flood, as opposed to depth and timing of flood propagation) is governed by the aerial imagery products available (which only captures the extent of the flood, at a singular point in time). By electing this form of evaluation, we only analyze the strengths of NWM-HAND simulations at the given time-step coinciding with the time of image capture (not necessarily peak flooding).*

**Bring discussion of limitations and realistic potential up from the discussion to the introduction/abstract.**

Thanks for this important point suggestion. We have moved the ideas as suggested. Please see lines 48-51.

*The current objective of the NWM-HAND approach is rapid flood prediction for the purposes of disaster warning and guidance. Model accuracy should therefore be viewed in this context and expectations should be tempered while recognizing the importance of having an operational, continental scale flood forecasting system.*

**Dr. Blodgett asked us to include more information regarding the NWM forcing data, parameterization, and routing.**

Thank you for this suggestion. To be upfront, deciding on the level of detail to include with respect to NWM, HAND, SRC, and USFIMR background has proven to be a difficult task to making this paper both complete and concise. As such we have done two things. (1) made an explicit data section in the background. (2) We listed the attributes of the model you suggest but have avoided discussing any implications. Further, we point readers to a presentation talking about the model in detail (please see lines 120-128)

**Dr. Blodgett noted that the introduction is lacking a general overview of the NWM's objectives which could/should be used to temper the expectations and focus the aims of an evaluation.**

Thank you for pointing this out. We have added a caution of sorts to the introduction as well as a statement reflecting the current goals and objectives of the model (please see lines 48-51; see above)

**Noting that the NWM-HAND system is not used for official forecasts and is to be considered for guidance only at this stage in its development. Given these kinds of caveats, the evaluation presented in this manuscript is of great value as it demonstrates that the NWM-HAND system is producing flood inundation products that would be generally useful for the intended purpose.**

Thank you for this pointer and notes on caveats. By noting your suggested caveats, we think our discussion has become better focused. That said with the re-evaluation we now find the simulated inundation products are limited in accuracy when it comes to pointing out flooded extents.

**Dr. Blodgett requested more information on the nature of the retrospective model run noting that it is only calibrated in some locations and should not be expected to produce realistic flow volumes. Additionally, he observed that the retrospective does not assimilate observed streamflow and suggested including such a remark.**

Thank you for the comment, both of these have been noted (please see lines 130-134).

*Complimenting the operational products are 23-year reanalysis studies for NWM versions 1.0, 1.2, and 2.0. These products use downscaled NLDAS-2 climate forcing's with the standard NWM configuration. Unlike the operational Analysis and Assimilation product however, the reanalysis products do not assimilate observed streamflow and have been calibrated in limited number of basins (Gochis, 2016)*

**Dr. Blodgett asked why we had not included NHD Areas in our masked-out regions.**

Thank you for this suggestion. We were unaware of the NHD area product and have now included it in our mask. Moreover, the NHD Fcodes (for both water bodies and areas) have been listed in text to increase transparency (see lines 168-169).

*For each event, a waterbody mask was created by combining the perennial NHD water bodies (NHD Fcode 39004, 39009) and NHDAreas (NHD FCode 40300, 40307, 40308, 40309) in each extent.*

**Dr. Blodgett aske to re consider our binning by stream order?**

Despite variable density of streams across the country there was clear evidence in our evaluations that lower order reaches underpredicted flood extents while higher order reaches preformed better. We attribute this to the use of a single default Manning n coefficient in the SRC generation and discuss the implication of this in a few spots throughout the manuscript. Most relevantly, we show how stream order is a driving factor resulting in competing results seen at the floodplain and catchment analysis resolution.

**Dr. Blodgett suggested generating a driving hypothesis and provided the example that "Given that HAND is not a physically based model in that it does not route flow over the landscape or preserve mass, we would expect small errors in stage to produce large errors in inundated areas in low-relief landscapes."**

Thank you for this succinct explanation of the phenomenon we were trying to describe as "volume control" in regions of low-relief. This wording has been added to lines 396-398 and help clarify our point.

*Since HAND is not a physical model, it is unable to conserve volume through space or time. In areas of low relief, where many cells have similar if not equal HAND values, small errors in stage can have disproportionate errors in inundation extent at the 10m grid cell resolution.*

**Dr. Blodgett asked us to re-think the distinction of 'Errors in the NHD' as a section heading**

This point is greatly appreciated.  Paragraph 2 in section 4.3.1 now starts:

"*With respect to the streamlines it is important to recognize that the NHD was developed as a cartographic representation of the nation's waterways and using a cartographic toolset for hydrologic modelling and routing applications has inherent limitations.*

And discusses the previously listed issues in this context. A new paragraph about the challenges with using a cartographic data as a modelling geofabric has been made in lines 364-377 with specific references to issues of refactoring catchment delineations to more compact and consistent modelling units. More over the section heading has been changed to better represent this and add a brief discussion of DEM resolution.

*"Data Models: Use, Limitations, and Adaptions"*

Again, thank you to both reviewers for helping make this paper substantially better than its original submission,

Sincerely,

Mike Johnson, Dinuke Munasinghe, Dami Eyelade, Sagy Cohen

**References:**

Zheng, X., Maidment, D. R., Tarboton, D. G., Liu, Y. Y., & Passalacqua, P. (2018). GeoFlood: Large-Scale Flood Inundation Mapping Based on High-Resolution Terrain Analysis. Water Resources Research, 54(12), 10-013.

Cohen, S., G. R. Brakenridge, A. Kettner, B. Bates, J. Nelson, R. McDonald, Y. Huang, D. Munasinghe, and J. Zhang (2018), Estimating Floodwater Depths from Flood Inundation Maps and Topography, *Journal of the American Water Resources Association*, 54 (4), 847–858.

---

## Author Response (AR2)

Dear Dr. Kettner,

Thank you for your feedback and suggestions. In the attached manuscript the following changes have been.

**Lines 57-59 now read:** *"In this paper we evaluate the integrated skill of NWM-HAND for simulating riverine flooding events by comparing NWM-HAND forecasts against an observational dataset at two scales, the floodplain, and the reach-level catchment."*

**Line 317:** "*that*" was removed

**Lines 364-365:** "*order*" was added

**Old lines 574-576 now read**: *"In every cross-sectional geometry we tested, observed streamflow (NWIS) / stage (USFIMR) relationships could be achieved by varying N except in catchments with zero relief (e.g. Figure 7B)."*

We have also added titles to figure 2 and 4 and improved the resolution of all images.

Thank you again,

Mike